# BI-LORA: EFFICIENT SHARPNESS-AWARE MINIMIZATION FOR FINE-TUNING LARGE-SCALE MODELS

**Yuhang Liu**[1,*] **Tao Li**[1,*], **Zhehao Huang**[1], **Zuopeng Yang**[1], **Xiaolin Huang**[1,2,†]
[1]Institute of Image Processing and Pattern Recognition, School of Automation and Intelligent Sensing, Shanghai Jiao Tong University
[2]MoE Key Laboratory of System Control and Information Processing (Shanghai)
`{yuhangliu,li.tao,kinght_h,yzpeng,xiaolinhuang}@sjtu.edu.cn`

## ABSTRACT

Low-Rank Adaptation (LoRA) enables parameter-efficient fine-tuning of large pre-trained models. Yet LoRA can face generalization challenges. One promising way to improve the generalization is Sharpness-Aware Minimization (SAM), which has proven effective for small-scale training scenarios. In this paper, we propose **Bi**-directional **Lo**w-**R**ank **A**daptation (Bi-LoRA), which introduces an auxiliary adversarial LoRA module. This design explicitly decouples sharpness optimization, handled by the auxiliary module, from task adaptation, performed by the primary module. Such a separation yields two key benefits. First, it transforms SAM's sequential computation of adversarial perturbation and gradient descent into a parallel form, which roughly halves the time and conquers the main obstacle of applying SAM in LoRA. Second, it provides perturbations from the auxiliary module that do not collapse into the restricted optimization subspace of the primary module, enabling broader sharpness exploration and flatter minima. Bi-LoRA simultaneously achieves both efficiency and effectiveness within a single framework, as validated by extensive experiments across diverse architectures and tasks. Code is available at `https://github.com/CrazyElements/Bi-LoRA`.

## 1 INTRODUCTION

The paradigm of pretraining followed by fine-tuning has become the de facto standard in machine learning, demonstrating state-of-the-art performance across various tasks (Devlin et al., 2018; Kolesnikov et al., 2020; Dosovitskiy et al., 2021; Radford et al., 2021). However, as model sizes continue to grow, full fine-tuning (Full FT) becomes memory-prohibitive in resource-constrained settings. The most successful approach for reducing memory cost in fine-tuning large-scale model is Low-Rank Adaptation (LoRA) (Hu et al., 2022), which introduces trainable, task-specific low-rank matrices to model weight updates. LoRA greatly lowers memory requirements by significantly reducing the trainable parameters and has become one of the most popular solutions for fine-tuning and deploying large models due to its simplicity and performance comparable to Full FT.

Despite LoRA's memory efficiency, it still faces generalization challenges when fine-tuned with limited data. In such scenarios, the risk of overfitting becomes particularly acute and can significantly compromise model performance (Li et al., 2025; Deng et al., 2025; Lin et al., 2024; Li et al., 2024b).

Motivated by the connection between flatness of the loss landscape and generalization, one promising direction for improving generalization is to seek flat minima (Hochreiter & Schmidhuber, 1994; 1997). Sharpness-Aware Minimization (SAM) (Foret et al., 2021) is a widely used technique that enhances generalization by formulating optimization as a min-max problem, effectively minimizing the worst-case loss within a local neighborhood. It has achieved state-of-the-art performance in small-scale training scenarios (Chen et al., 2022; Zhuang et al., 2022). However, SAM requires calculating an adversarial perturbation of model parameters at each training step, incurring additional memory overhead of a copy of the model weights and doubling the training time. When applied to large-scale models, such extra memory cost and computation becomes particularly pronounced.

---

*Equal Contribution.
†Corresponding Author.

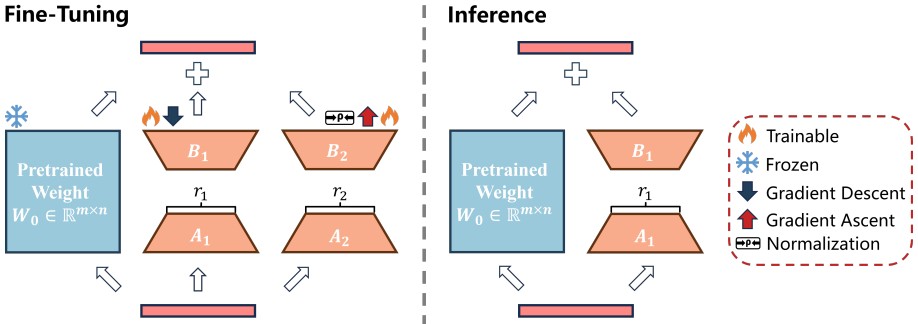

Figure 1: Overview of our proposed Bi-LoRA. During fine-tuning (**Left**), Bi-LoRA utilizes two opposing LoRA modules: the primary module ($B_1A_1$) is optimized by conventional gradient descent for task-specific adaptation, while the auxiliary module ($B_2A_2$) performs gradient ascent for sharpness optimization. These two modules are decoupled and simultaneously optimized. During inference (**Right**), only ($B_1A_1$) is retained and merged with the pretrained weights.

Thus, a natural approach to finding flat minima when fine-tuning large models, without sacrificing memory efficiency, is to apply SAM to the LoRA parameters, referred to as LoRA-SAM (Li et al., 2024a). However, this straightforward integration still suffers from the doubled training cost of SAM. To address this essential problem, we introduce an auxiliary adversarial LoRA module to decouple the sharpness optimization from the task adaptation. This design, Bi-directional Low-Rank Adaptation (Bi-LoRA) shown in Figure 1, enables updating both modules in one forward and backward pass, making SAM practical for fine-tuning large-scale models.

Decoupling the two LoRA modules also enables broader exploration of perturbations. During inference, the LoRA parameters are merged into the pretrained weights, so we should care about the sharpness of the full parameter space. In contrast, LoRA-SAM limits adversarial perturbations to a restricted subspace (see Proposition 1). Worse, LoRA-SAM collapses quickly into it (see Figure 2c), thereby optimizing sharpness only there. Empirically, Bi-LoRA alleviates this issue: its decoupled auxiliary perturbation space converges more slowly than the primary optimization space, enabling exploring perturbation beyond the restricted subspace to capture sharpness and promote flatter minima. Bi-LoRA maintains LoRA's efficiency while improving generalization, making it a strong alternative for fine-tuning large-scale models.

Our contributions can be summarized as follows:

- We propose **Bi-LoRA**, which introduces an auxiliary LoRA module to model SAM's adversarial weight perturbation, decoupling it from LoRA optimization. This design enables simultaneous optimization of SAM's two steps with minor additional memory costs.
- We point out that directly applying SAM to LoRA parameters can only optimize the sharpness within the restricted subspace, thereby limiting its potential to improve generalization. Bi-LoRA broadens the sharpness exploration: the auxiliary perturbation module converges more slowly than the primary one, enabling exploration beyond the restricted subspace.
- Extensive experiments across a wide range of fine-tuning tasks, including natural language understanding, mathematics, code generation, chat, instruction following, and Text-to-Image generation, demonstrate that Bi-LoRA achieves superior generalization performance.

## 2 ISSUES OF LoRA-SAM

In this section, we formalize LoRA-SAM and analyze its subspace collapse issue during training. Related work on LoRA and SAM is summarized in Appendix E.

### 2.1 PRELIMINARIES

Given a pre-trained weight matrix $W_0 \in \mathbb{R}^{m \times n}$, LoRA utilizes low-rank matrices $B$ and $A$ to model the weight change $\Delta W$ during the fine-tuning,

$$W = W_0 + \Delta W \approx W_0 + BA, \tag{1}$$

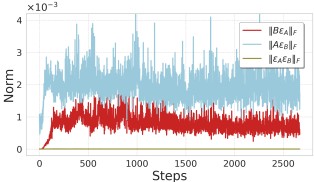 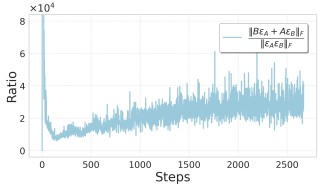 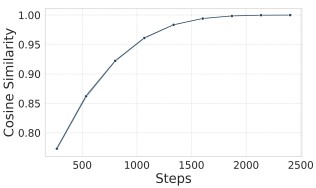

(a) Norm of different components     (b) Norm ratio     (c) Cosine similarity to final

Figure 2: Training statistics for LoRA-SAM during fine-tuning. (**a**) and (**b**): Frobenius norms of different terms and the ratio of the Frobenius norms of $(B\epsilon_A + \epsilon_B A)$ to that of $(\epsilon_B \epsilon_A)$ in Eqn. (5). We monitor a fixed LoRA module and more similar experiments can be found in Appendix T. It could be observed that the norm of the third term ($\epsilon_B \epsilon_A$) is several orders of magnitude ($> 10^4$) smaller than that of the first two terms, making it negligible. Note that $\|B\epsilon_A\|_F$ is initially zero since $B$ are initialized to zero by default (Hu et al., 2022). (**c**): Cosine similarity between the final LoRA parameters and those during fine-tuning. The LoRA parameters converge rapidly during fine-tuning. The experiments are conducted on CoLA with the T5-base model.

where $B \in \mathbb{R}^{m \times r}$ and $A \in \mathbb{R}^{r \times n}$ with $r \ll \min\{m, n\}$. Here we omit the scaling factor $s = \alpha/r$ for the sake of simplicity in the equation, as it can be easily incorporated into $B$ and $A$.

During gradient back-propagation, the loss gradient w.r.t. $B$ and $A$ is computed using the chain rule:

$$\frac{\partial \mathcal{L}}{\partial B} = (\nabla_W \mathcal{L})A^\top, \quad \frac{\partial \mathcal{L}}{\partial A} = B^\top (\nabla_W \mathcal{L}), \tag{2}$$

where $\mathcal{L}$ is the loss objective to be minimized.

## 2.2 LoRA-SAM

For optimizing the sharpness while keeping memory efficiency, a natural idea is to apply SAM over the LoRA parameters (LoRA-SAM) which formulates the following optimization target:

$$\min_{B,A} \max_{\|(\epsilon_B, \epsilon_A)\| \leq \rho} \mathcal{L}\left(W_0 + (B + \epsilon_B)(A + \epsilon_A)\right), \tag{3}$$

where $\epsilon_B \in \mathbb{R}^{m \times r}, \epsilon_A \in \mathbb{R}^{r \times n}$ are the adversarial weight perturbations over low-rank matrices, $\|(\epsilon_B, \epsilon_A)\|$ denotes the norm of weight perturbations (a typical setting is the $\ell_2$-norm), and $\rho$ is the neighborhood radius.

To efficiently solve the inner maximization problem in Eqn. (3), Foret et al. (2021) employ first-order Taylor expansion for approximation, and the resulting perturbations are calculated as follows:

$$\epsilon_B = \rho \cdot \frac{\partial \mathcal{L}}{\partial B} / F_{\text{total}}, \quad \epsilon_A = \rho \cdot \frac{\partial \mathcal{L}}{\partial A} / F_{\text{total}}, \quad F_{\text{total}} = \sqrt{\left\|\frac{\partial \mathcal{L}}{\partial B}\right\|_F^2 + \left\|\frac{\partial \mathcal{L}}{\partial A}\right\|_F^2}, \tag{4}$$

where $\|\cdot\|_F$ denotes the Frobenius norm. Then we can rewrite the perturbation given by Eqn. (3) as

$$\epsilon_W = B\epsilon_A + \epsilon_B A + \epsilon_B \epsilon_A. \tag{5}$$

Since both $\epsilon_A$ and $\epsilon_B$ are several orders of magnitude smaller than the original matrices $A$ and $B$, the cross term $\epsilon_B \epsilon_A$ is negligible, as evidenced by Figures 2a and 2b, which report the magnitudes of these terms. Therefore, Eqn. (5) can be approximately simplified as:

$$\epsilon_W \approx c \left[ BB^\top (\nabla_W \mathcal{L}) + (\nabla_W \mathcal{L})A^\top A \right], \tag{6}$$

where $c = \rho / \sqrt{\|(\nabla_W \mathcal{L})A^\top\|_F^2 + \|B^\top(\nabla_W \mathcal{L})\|_F^2}$. Now one can observe that the perturbation of LoRA-SAM is confined to a restricted subspace, as demonstrated below:

**Proposition 1** (Perturbation Space of LoRA-SAM). *The effective weight perturbation in LoRA-SAM can be decomposed into two terms: $BB^\top(\nabla_W \mathcal{L})$ and $(\nabla_W \mathcal{L})A^\top A$. The column space of the first term is given by $Col(B)$, while the row space of the second term is given by $Row(A)$.*

The space for adversarial weight perturbation in LoRA-SAM is primarily dominated by $Col(B)$ and $Row(A)$, from which it follows that LoRA-SAM only cares about the sharpness within the subspace defined by $B$ and $A$, failing to capture the sharpness in a broader space.

Moreover, $B$ and $A$ converge rapidly during training, as shown in Figure 2c. **This leads to the fast convergence of the column and row spaces** $\text{Col}(B)$ and $\text{Row}(A)$, **further restricting the subspace for sharpness optimization and hindering the effectiveness of LoRA-SAM**, potentially leading to suboptimal performance. Appendix U further corroborates this subspace collapse phenomenon.

In Figure 4, we compare the loss landscape flatness of different methods. We observe that LoRA-SAM achieves the flattest loss landscape within the LoRA parameter space as in Figure 4a, reflecting its original sharpness-minimization objective. In contrast, Figure 4b shows that while LoRA-SAM yields a flatter landscape than LoRA in the full space, the loss still rises sharply with perturbation magnitude. This indicates the limitation of applying perturbation in a restricted subspace, highlighting the importance of considering sharpness over a broader space beyond LoRA subspace.

## 3 BI-LoRA: BI-DIRECTIONAL LOW-RANK ADAPTATION

In this section, we introduce Bi-LoRA, a novel LoRA variant, which contains an auxiliary module beyond the primary module in regular LoRA. The aim of the auxiliary module is to decouple the sharpness optimization from task adaptation, which makes the training efficient and enhances the generalization improvement. Specifically, the proposed Bi-LoRA takes the following formulation,

$$W = W_0 + B_1 A_1 + B_2 A_2, \tag{7}$$

where the first LoRA module $(A_1, B_1)$ serves as the primary module responsible for task-specific adaptation, similar to standard LoRA, while the second module $(A_2, B_2)$ acts as the auxiliary LoRA module for modeling adversarial perturbation in SAM. With these two modules, the Bi-LoRA's optimization objective is given below,

$$\min_{B_1, A_1} \max_{\|B_2 A_2\|_F \leq \rho} \mathcal{L}\left(W_0 + B_1 A_1 + B_2 A_2\right), \tag{8}$$

where $\rho > 0$ is the neighborhood radius that controls the magnitude of perturbations as in the original SAM. After training, we discard the auxiliary modules $(A_2, B_2)$, as they serve solely to optimize sharpness during training and guide the primary module $(A_1, B_1)$ towards a flat region. Consequently, the adapted weights are reduced to the primary module, i.e., $W = W_0 + B_1 A_1$, preserving the original LoRA structure and ensuring no additional computational overhead during inference.

In LoRA-SAM, the adversarial perturbation is calculated as Eqn. (4), where the backpropagation on $(\epsilon_A, \epsilon_B)$ and on $(A, B)$ must be carried out sequentially. Now in Bi-LoRA, the task adaptation and the perturbation are decoupled, as shown in Eqn. (8). Then one can simultaneously perform task adaptation and sharpness-aware minimization *in one gradient step*. Specifically, during each iteration, the primary LoRA module $(A_1, B_1)$ is updated through standard gradient descent, while the auxiliary LoRA module $(A_2, B_2)$ is updated through gradient ascent for sharpness optimization.

The bi-directional gradient update approach embodies the core concept of our Bi-LoRA. Concretely, one update step for Bi-LoRA is formalized as follows:

$$\begin{cases} B_1^{k+1} = B_1^k - \eta_1 \left(\nabla_W \mathcal{L}\right) A_1^{k\top}, & A_1^{k+1} = A_1^k - \eta_1 B_1^{k\top} \left(\nabla_W \mathcal{L}\right), \\ B_2^{k+1} = B_2^k + \eta_2 \left(\nabla_W \mathcal{L}\right) A_2^{k\top}, & A_2^{k+1} = A_2^k + \eta_2 B_2^{k\top} \left(\nabla_W \mathcal{L}\right), \end{cases} \tag{9}$$

where $\nabla_W \mathcal{L} = \left.\frac{\partial \mathcal{L}}{\partial W}\right|_{W=W_0+B_1^k A_1^k + B_2^k A_2^k}$ is the gradient of the loss $\mathcal{L}$ w.r.t. the merged weight $W$, $k$ denotes the iteration index, and $\eta_1, \eta_2$ are learning rates. We use the same learning rate for both LoRA modules in our experiments, though other choices are possible (see Appendix Q).

The independent backpropagation for the two LoRA modules $(A_1, B_1)$ and $(A_2, B_2)$ makes the updates parallelizable. This eliminates the doubled computational cost typically incurred by LoRA-SAM. The following proposition ensures that the adversarial direction induced by $(A_2, B_2)$ can still increase the inner objective, in non-negatively alignment with the SAM's perturbation direction. The Proof and further discussions are provided in Appendix F.1.

**Proposition 2** (Alignment of Bi-LoRA's ascent direction with previous full gradient). *Let $G_t = \nabla_W \mathcal{L}(W_0 + B_{1,t} A_{1,t} + \tilde{\epsilon}_t)$ denote the full gradient at step $t$, with $\tilde{\epsilon}_t = B_{2,t} A_{2,t}$. After one Bi-LoRA update,*

$$\langle G_t, \tilde{\epsilon}_{t+1} - \tilde{\epsilon}_t \rangle \geq 0,$$

*i.e., Bi-LoRA increases the inner objective along the previous SAM's perturbation direction.*

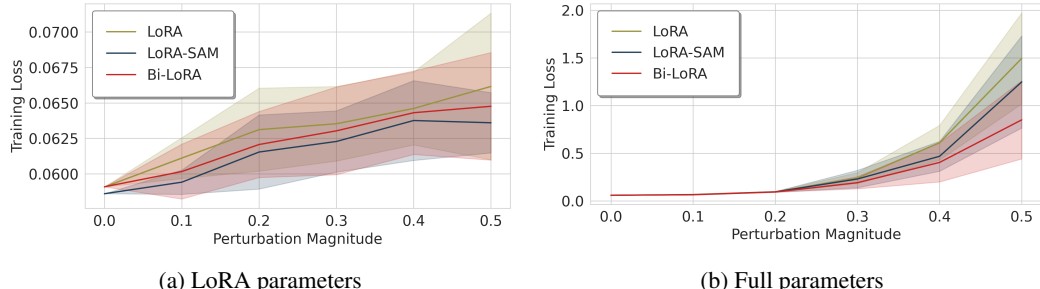

(a) LoRA parameters  (b) Full parameters

Figure 4: Loss landscape visualization along random "filter-normalized" directions following Li et al. (2018), focusing on (**a**) LoRA parameters and (**b**) full parameters. It can be observed that while LoRA-SAM attains the greatest flatness within the LoRA subspace, which aligns with its sharpness-minimization objective, it is not the flattest in the full parameter space. This distinction is critical for inference, as LoRA parameters are ultimately merged into the pretrained weights. In contrast, Bi-LoRA delivers a substantially greater improvement on flatness in the full parameter space. All experiments were averaged over five independent runs with T5-base fine-tuned on CoLA.

Decoupling sharpness optimization from task adaptation not only makes the backpropagation parallelizable but also partially alleviates the inconsistency of LoRA-SAM, where perturbations are applied to a restricted subspace $(A, B)$ while inference is determined by the full parameters (see Proposition 1). Bi-LoRA eliminates the dependence of adversarial weight perturbations on the primary LoRA module. In Eqn. 8, we see the column space for perturbations in Bi-LoRA is spanned by $\text{Col}(B_2)$, which is independent of the LoRA optimization space, i.e., $\text{Col}(B_1)$.

Although Bi-LoRA's perturbation space $\text{Col}(B_2)$ (auxiliary) remains a subspace, it is strictly decoupled from the optimization space $\text{Col}(B_1)$ (primary). Moreover, we find that the auxiliary modules converge *much more slowly* than the primary ones, preserving flexibility for sharpness-aware updates (see Figure 3 and

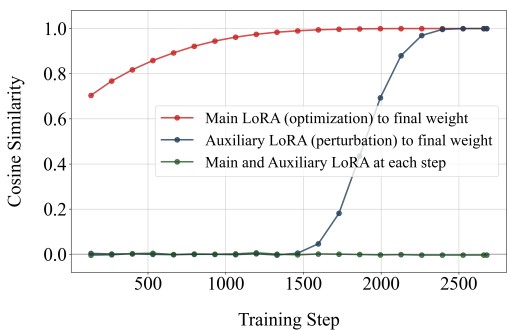

Figure 3: Cosine similarity between the main LoRA (task adaptation, red), the auxiliary LoRA (perturbation, blue), and their trajectories (green) during training T5-base on CoLA. The auxiliary converges only in the last 20% of steps, substantially slower and independent of the main.

Appendix L). Therefore, Bi-LoRA enables a more effective capture of sharpness in the full parameter space. And Figure 4 confirms that Bi-LoRA achieves a significantly flatter loss landscape in the full parameter space, leading to improved generalization.

Numerous studies have discussed SAM's convergence and generalization (Andriushchenko & Flammarion, 2022; Dai et al., 2023; Khanh et al., 2024). For Bi-LoRA, the following proposition establishes that the Bi-LoRA objective (Eqn. 8) is essentially equivalent to the vanilla LoRA optimization with this regularization term.

**Proposition 3** (Bi-LoRA is a Regularized LoRA). *From the gradient-norm perspective, SAM can be expressed as*

$$\mathcal{L}(W) + \rho \left\| \nabla_W \mathcal{L}(W) \right\|_2 .$$

*If the inner maximization of Bi-LoRA is solved to convergence and attains its optimum, its objective reduces to a low-rank counterpart:*

$$\min_{B_1, A_1} \mathcal{L}(W_0 + B_1 A_1) + \rho \left\| \nabla_{W_0 + B_1 A_1} \mathcal{L} \right\|_{(r)} ,$$

*where $\| \cdot \|_{(r)}$ denotes the Ky Fan $r$-norm (the sum of the top-$r$ singular values). Thus, ideally, Bi-LoRA can be viewed as LoRA equipped with an explicit low-rank gradient-norm regularizer.*

Proofs and further details are deferred to the Appendix F.2. Proposition 3 coincides with the regularization view of SAM (Yue et al., 2023). Bi-LoRA can be analyzed in the same way, differing

---

**Algorithm 1** Bi-LoRA

---

1: **Input:** Initial weight $W_0$, learning rates $\eta_1$, $\eta_2$, radius $\rho$, number of LoRA layers $N$
2: **Output:** Adapted weight $W$ for inference
3: Initialize LoRA modules $B_1^0$, $A_1^0$, $B_2^0$, $A_2^0$;
4: $k \leftarrow 0$;
5: **while** *not converged* **do**
6:    Sample mini-batch data $\mathcal{B}$;
7:    Apply gradient descent to primary and ascent to auxiliary LoRA modules via Eqn. (9);
8:    Clip the auxiliary modules via Eqn. (10);
9:    $k \leftarrow k + 1$;
10: **end while**
11: Remove the auxiliary LoRA modules $B_2^k$, $A_2^k$;
12: **return** $W^k = W_0 + B_1^k A_1^k$

---

only in that its regularizer is a low-rank norm. In practice, however, we perform a single inner step to preserve high efficiency, which is crucial for fine-tuning models. This choice introduces a gap between the practical Bi-LoRA and analyses that require optimality of the inner optimization. Given our application-oriented focus, we do not delve deeply into detailed theoretical discussions.

**Norm Constraints.**   In Eqn. (8), we impose a norm constraint to keep the perturbation sufficiently small to avoid disrupting normal model training. Thus, we apply global clipping to the auxiliary module $(A_2, B_2)$ after each update, by constraining their total Frobenius norm, which maintains the weight perturbation within a controlled magnitude. Specifically, suppose that there are $N$ LoRA layers. The $i$-th auxiliary module is scaled as follows:

$$\begin{cases} B_2^{(i)} \leftarrow \sqrt{\rho/c_{\text{norm}}} \cdot B_2^{(i)}, \\ A_2^{(i)} \leftarrow \sqrt{\rho/c_{\text{norm}}} \cdot A_2^{(i)}, \end{cases} \quad \text{if} \quad c_{\text{norm}} > \rho. \tag{10}$$

Notice the normalization is applied only if the total Frobenius norm over $N$ auxiliary LoRA modules $c_{\text{norm}}$ exceeds the neighborhood radius $\rho$, where $c_{\text{norm}} = \sqrt{\sum_{j=1}^{N} \|B_2^{(j)} A_2^{(j)}\|_F^2}$. This ensures the perturbation remains constrained within the preset $\rho$-norm ball. The overall procedure for Bi-LoRA is summarized in Algorithm 1.

## 4 EXPERIMENTS

In this section, we evaluate Bi-LoRA across diverse models and benchmark tasks. We test Bi-LoRA's capacities on: (1) Llama 2/3.1 models (Touvron et al., 2023; Dubey et al., 2024) for mathematical reasoning, coding, dialogue, and instruction following; (2) Qwen 2.5-14B (Qwen et al., 2025), a larger backbone for instruction-following; and (3) SDXL for text-to-image generation. We further demonstrate that Bi-LoRA can be integrated with existing LoRA variants to provide consistent improvement, and conduct ablation studies to examine its hyperparameter sensitivity. More results on natural language understanding tasks with T5-base are provided in Appendix I.

### 4.1 BASELINES

We mainly compare with the following baseline methods:

- **Full FT** fine-tunes all model parameters.
- **LoRA** applies low-rank adaptation to all linear modules.
- **LoRA-SAM** applies sharpness-aware minimization over LoRA parameters.
- **Random perturbation baselines:** Random Output Perturbation (ROP) perturbs logits $y$ with Gaussian noise $\epsilon$, i.e., $\tilde{y} = y + \epsilon$, $\epsilon \sim \mathcal{N}(0, \rho^2 I)$; Random Weight Perturbation (RWP)_full/LoRA perturb either full weights or only LoRA adapters with Gaussian noise.
- **SAM variants**, including LoRA-oBAR/nBAR (Li et al., 2024a), Flat-LoRA (Li et al., 2025), LoRA-ESAM (Du et al., 2022a), LoRA-LookSAM (Liu et al., 2022b), $S^2$ SAM (Ji et al., 2024), and WSAM (Yue et al., 2023).
- **LoRA variants**, including LoRA-GA (Wang et al., 2024), PiSSA (Meng et al., 2024), DoRA (Liu et al., 2024), HiRA (Huang et al., 2025), and DeLoRA (Bini et al., 2025).

Table 1: Results of fine-tuning Llama 2-7B and Llama 3.1-8B on different tasks. "Cost" indicates the gradient steps per training iteration, e.g., one step (Cost $\times 1$) for Full FT, LoRA, and Bi-LoRA.

| Method | Cost | Llama 2-7B | | | Llama 3.1-8B | | | |
| --- | --- | --- | --- | --- | --- | --- | --- | --- |
| | | GSM8K | HumanEval | MT-Bench | MMLU | DROP | HEval | BBH |
| Full FT | $\times 1$ | $59.74_{\pm 0.69}$ | $33.12_{\pm 0.32}$ | $6.16_{\pm 0.09}$ | $64.31_{\pm 0.31}$ | $51.52_{\pm 0.45}$ | $41.45_{\pm 1.58}$ | $44.78_{\pm 0.33}$ |
| LoRA | $\times 1$ | $58.21_{\pm 0.34}$ | $24.75_{\pm 0.23}$ | $5.92_{\pm 0.10}$ | $63.38_{\pm 0.39}$ | $49.82_{\pm 0.54}$ | $43.15_{\pm 0.93}$ | $42.82_{\pm 0.27}$ |
| LoRA-SAM | $\times 2$ | $59.16_{\pm 0.52}$ | $26.59_{\pm 0.36}$ | $5.97_{\pm 0.08}$ | $63.46_{\pm 0.19}$ | $50.94_{\pm 0.22}$ | $44.36_{\pm 1.13}$ | $43.49_{\pm 0.40}$ |
| ROP | $\times 1$ | $59.24_{\pm 0.70}$ | $25.41_{\pm 0.11}$ | $6.05_{\pm 0.08}$ | $63.63_{\pm 0.20}$ | $49.96_{\pm 0.26}$ | $42.27_{\pm 0.81}$ | $43.47_{\pm 0.29}$ |
| RWP_full | $\times 1$ | $59.41_{\pm 0.59}$ | $26.26_{\pm 0.35}$ | $6.01_{\pm 0.12}$ | $63.40_{\pm 0.14}$ | $50.11_{\pm 0.12}$ | $44.31_{\pm 0.81}$ | $43.35_{\pm 0.18}$ |
| RWP_LoRA | $\times 1$ | $58.81_{\pm 0.27}$ | $24.92_{\pm 0.11}$ | $5.80_{\pm 0.11}$ | $63.50_{\pm 0.01}$ | $50.16_{\pm 0.67}$ | $42.68_{\pm 0.70}$ | $\mathbf{44.10}_{\pm 0.25}$ |
| LoRA-oBAR | $\times 1$ | $59.26_{\pm 0.53}$ | $26.30_{\pm 0.33}$ | $5.97_{\pm 0.07}$ | $63.62_{\pm 0.12}$ | $49.92_{\pm 0.48}$ | $43.49_{\pm 0.20}$ | $43.44_{\pm 0.12}$ |
| LoRA-nBAR | $\times 1$ | $59.72_{\pm 0.25}$ | $26.50_{\pm 0.23}$ | $6.10_{\pm 0.06}$ | $63.45_{\pm 0.10}$ | $49.80_{\pm 0.03}$ | $45.23_{\pm 0.20}$ | $43.39_{\pm 0.17}$ |
| Flat-LoRA | $\times 1$ | $59.44_{\pm 0.33}$ | $26.67_{\pm 0.23}$ | $5.98_{\pm 0.05}$ | $\mathbf{63.67}_{\pm 0.33}$ | $50.44_{\pm 0.17}$ | $44.31_{\pm 0.73}$ | $43.99_{\pm 0.10}$ |
| LoRA-ESAM | $\times 1.x$ | $58.33_{\pm 0.22}$ | $24.84_{\pm 0.04}$ | $5.80_{\pm 0.13}$ | $61.79_{\pm 1.65}$ | $49.31_{\pm 0.30}$ | $42.48_{\pm 1.33}$ | $43.40_{\pm 0.15}$ |
| LoRA-LookSAM | $\times 1.x$ | $58.55_{\pm 0.28}$ | $25.28_{\pm 0.29}$ | $5.94_{\pm 0.08}$ | $63.45_{\pm 0.29}$ | $50.34_{\pm 0.34}$ | $42.48_{\pm 0.54}$ | $43.28_{\pm 0.20}$ |
| Bi-LoRA | $\times 1$ | $\mathbf{60.32}_{\pm 0.30}$ | $\mathbf{27.20}_{\pm 0.42}$ | $\mathbf{6.26}_{\pm 0.06}$ | $\mathbf{63.67}_{\pm 0.15}$ | $\mathbf{51.53}_{\pm 0.33}$ | $\mathbf{46.12}_{\pm 0.89}$ | $43.45_{\pm 0.31}$ |

To more rigorously evaluate the effectiveness in improving generalization, we adopt a stronger training protocol with larger learning rate than prior works (Wang et al., 2025; 2024). The same protocol is applied across all methods for fairness. Unless otherwise specified, all results are averaged over three independent runs with standard errors. See Appendix R for detailed hyperparameter settings and more training protocol.

## 4.2 RESULTS ON LLAMA MODELS

**Setting.** We evaluate the performance of Bi-LoRA on Llama 2-7B/3.1-8B across four tasks: mathematical reasoning, code generation, dialogue generation, and instruction following, following Wang et al. (2024); Li et al. (2025); Ren et al. (2024). Llama 2-7B is fine-tuned on the first three tasks, while Llama 3.1-8B is used for instruction following. Each task focuses on a specific capability and uses well-established datasets and metrics for training and evaluation, as detailed below:

- **Mathematical reasoning.** Our model is fine-tuned on a 100k subset of MetaMathQA and evaluated on GSM8K. Performance is measured by accuracy.
- **Code generation.** We fine-tune our model on a 100k subset of Code-Feedback and evaluate it on the HumanEval benchmark, using the PASS@1 metric.
- **Dialogue generation.** We train our model on the WizardLM dataset and evaluate it on MT-Bench. The response quality is assessed with GPT-4, and we report the first-turn score on a 10-point scale.
- **Instruction following.** Fine-tuned on Cleaned Alpaca (Taori et al., 2023) and evaluated on INSTRUCTEVAL (Chia et al., 2023), reporting exact match for MMLU, DROP, and BBH, and PASS@1 for HumanEval.

In our experiments, following Du et al. (2022a), (Liu et al., 2022b), LoRA-ESAM perturbs 50% of the parameters on the top-50% sharpness-sensitive data, while LoRA-LookSAM applies SAM perturbation every five steps, and thus their costs are denoted as "$\times 1.x$".

**Results.** We begin with Llama 2-7B for math, code, and chat tasks. The results, presented in Table 1 (**left**), demonstrate Bi-LoRA's superior performance. Compared to LoRA, Bi-LoRA achieves improvements of 2.11% accuracy on GSM8K, 2.45% on HumanEval, and 0.34 on MT-Bench. Importantly, these gains are even more pronounced than those achieved with the smaller T5-base model (Appendix I), and Bi-LoRA operates at a similar speed as LoRA, underscoring its scalability. Furthermore, Bi-LoRA narrows the gap between LoRA fine-tuning and full fine-tuning, and notably outperforms full fine-tuning on GSM8K and MT-Bench tasks, which are not attainable by competing methods. In contrast, we observe that LoRA-SAM does not consistently improve upon LoRA. Random-perturbation baselines provide only marginal or task-specific gains relative to LoRA-SAM, showing that trivial random noise perturbation offers limited generalization improvement. Unlike efficient-SAM variants that either incur extra cost (LoRA-ESAM, LoRA-LookSAM) or compromise performance on certain tasks (e.g., Flat-LoRA on DROP), Bi-LoRA achieves consistent gains on all benchmarks at a single-step computation.

Next, we turn to the instruction following task with Llama 3.1-8B. As shown in Table 1 (**right**), Bi-LoRA outperforms almost all baselines, surpassing LoRA by 0.29% on MMLU, 1.71% on DROP,

Table 2: Results of fine-tuning Qwen 2.5-14B on instruction-following tasks. "Cost" indicates the gradient steps per training iteration, e.g., one step (Cost $\times 1$) for LoRA and Bi-LoRA.

| Method | Cost | MMLU | DROP | HEval | BBH | Avg. |
|---|---|---|---|---|---|---|
| Vanilla | $\times 1$ | $79.28_{\pm 0.17}$ | $51.55_{\pm 0.76}$ | $70.93_{\pm 1.13}$ | $57.47_{\pm 0.03}$ | 64.81 |
| LoRA-SAM | $\times 2$ | $79.19_{\pm 0.31}$ | $54.54_{\pm 1.03}$ | $70.53_{\pm 1.81}$ | $58.25_{\pm 0.11}$ | 65.63 |
| ROP | $\times 1$ | $79.22_{\pm 0.28}$ | $52.74_{\pm 1.42}$ | $71.34_{\pm 0.35}$ | $57.84_{\pm 0.49}$ | 65.28 |
| RWP_full | $\times 1$ | $78.92_{\pm 0.07}$ | $54.75_{\pm 1.01}$ | $71.13_{\pm 1.47}$ | $57.76_{\pm 0.07}$ | 65.64 |
| RWP_LoRA | $\times 1$ | $79.07_{\pm 0.22}$ | $51.69_{\pm 0.71}$ | $71.34_{\pm 0.93}$ | $57.45_{\pm 0.22}$ | 64.89 |
| LoRA-oBAR | $\times 1$ | $79.37_{\pm 0.28}$ | $55.70_{\pm 0.61}$ | $69.92_{\pm 0.20}$ | $58.53_{\pm 0.46}$ | 65.88 |
| LoRA-nBAR | $\times 1$ | $79.34_{\pm 0.32}$ | $55.96_{\pm 0.45}$ | $69.92_{\pm 0.20}$ | $58.67_{\pm 0.44}$ | 65.97 |
| Flat-LoRA | $\times 1$ | $79.51_{\pm 0.20}$ | $54.90_{\pm 1.06}$ | $\mathbf{71.54}_{\pm 0.20}$ | $58.13_{\pm 0.45}$ | 66.02 |
| LoRA-ESAM | $\times 1.x$ | $79.55_{\pm 0.03}$ | $54.83_{\pm 0.97}$ | $65.04_{\pm 0.81}$ | $58.43_{\pm 0.32}$ | 64.46 |
| LoRA-LookSAM | $\times 1.x$ | $79.63_{\pm 0.23}$ | $56.35_{\pm 1.55}$ | $67.88_{\pm 1.08}$ | $58.41_{\pm 0.07}$ | 65.57 |
| Bi-LoRA | $\times 1$ | $\mathbf{79.67}_{\pm 0.05}$ | $\mathbf{56.49}_{\pm 0.22}$ | $71.34_{\pm 0.93}$ | $\mathbf{58.93}_{\pm 0.05}$ | $\mathbf{66.61}$ |

2.97% on HumanEval, and 0.63% on BBH. While LoRA-SAM and the two types of baselines show improvements over LoRA, Bi-LoRA achieves more substantial gains, particularly on DROP and HumanEval datasets, while requiring only half the training time. Overall, Bi-LoRA offers a better accuracy and efficiency trade-off than both random perturbation and efficient SAM baselines.

### 4.3 RESULTS ON QWEN MODEL

To demonstrate Bi-LoRA's performance across model size and architecture, we next experiment on Qwen 2.5-14B, a distinct and larger backbone compared to LlaMA 2/3.1. We focus on instruction-following tasks. Table 2 shows that Bi-LoRA achieves the highest average score, improving over LoRA-SAM by 0.98%. Compared to the strongest alternative (Flat-LoRA) across both random perturbation and efficient SAM variants baselines, Bi-LoRA delivers an additional 0.59% average gain. This indicates its applicability across architectures and scales.

### 4.4 RESULTS ON DIFFUSION MODELS

**Setting.** We apply Bi-LoRA to a subject-driven generalization task and finetune SDXL (Podell et al., 2023) via Dreambooth (Ruiz et al., 2023) on 3D Icons dataset, which contains 23 square-icon images.

Table 3: CLIP I2T and T2T similarity (%) for LoRA and Bi-LoRA on the 3D Icons dataset.

| Method | CLIP I2T ($\uparrow$) | CLIP T2T ($\uparrow$) |
|---|---|---|
| LoRA | $32.43_{\pm 0.46}$ | $42.27_{\pm 2.08}$ |
| Bi-LoRA | $\mathbf{33.14}_{\pm 0.41}$ | $\mathbf{46.79}_{\pm 2.37}$ |

**Results.** Table 3 reports the average CLIP image-text (I2T) and text-text (T2T) similarity scores over six prompt instances of the form "*a ToK icon of a <Instance>, in the style of TOK*", each evaluated across 200 runs. Compared with LoRA, Bi-LoRA improves the average I2T and T2T similarities by 0.71 and 4.52, respectively, demonstrating its stronger personalization capability. As shown in Figure 5, in the second row, the image generated by Bi-LoRA artfully merges the fluffy rabbit with the icon, whereas LoRA either fails to integrate the rabbit into the icon (fourth) or even does not generate an icon (third). Furthermore, Bi-LoRA better preserves the attributes of the rabbit, such as the eyes in the first column. More qualitative examples are shown in Appendix K.

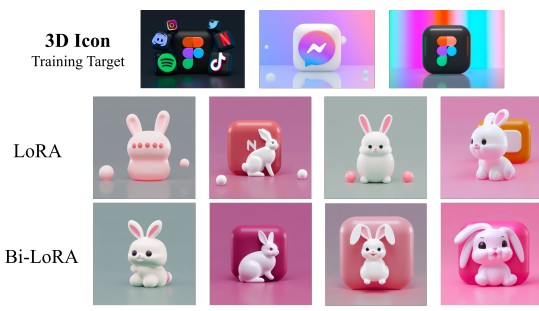

Prompt: *a ToK icon of a fluffy rabbit, in the style of ToK.*

Figure 5: Images generated by SDXL fine-tuned with LoRA and Bi-LoRA on the 3D icon datasets, where each column uses the *same* seed for fair comparisons.

Table 5: Ablations of each component in Bi-LoRA using Llama 3.1-8B on instruction following tasks. We compare vanilla LoRA, Dual-LoRA (two descent branches with global clipping), and Bi-LoRA under different clipping schemes (no clipping, per-layer clipping, and global clipping).

| Method | Clipping | Adversarial | MMLU | DROP | HEval | BBH | Avg. |
|---|---|---|---|---|---|---|---|
| LoRA | ✗ | ✗ | $63.38_{\pm 0.39}$ | $49.82_{\pm 0.54}$ | $43.15_{\pm 0.93}$ | $42.82_{\pm 0.27}$ | 49.79 |
| Dual-LoRA | global | ✗ | $63.49_{\pm 0.12}$ | $49.97_{\pm 0.17}$ | $42.71_{\pm 0.47}$ | $43.25_{\pm 0.15}$ | 49.86 |
| Bi-LoRA (no clip) | ✗ | ✓ | $61.33_{\pm 0.42}$ | $47.31_{\pm 3.66}$ | $40.24_{\pm 0.93}$ | $41.96_{\pm 0.48}$ | 47.71 |
| Bi-LoRA (per-layer clip) | per-layer | ✓ | $63.49_{\pm 0.26}$ | $50.26_{\pm 0.30}$ | $43.29_{\pm 0.61}$ | $43.06_{\pm 0.28}$ | 50.03 |
| **Bi-LoRA (global clip)** | global | ✓ | $\mathbf{63.67}_{\pm 0.15}$ | $\mathbf{51.53}_{\pm 0.33}$ | $\mathbf{46.12}_{\pm 0.89}$ | $\mathbf{43.45}_{\pm 0.15}$ | **51.19** |

## 4.5 INTEGRATION WITH OTHER LoRA VARIANTS

In this section, we evaluate Bi-LoRA's effectiveness when combined with existing advanced LoRA variants. Specifically, we consider three LoRA variants, including LoRA-GA (Wang et al., 2024), PiSSA (Meng et al., 2024) and DoRA (Liu et al., 2024), and fine-tune the T5-base model on the MRPC and CoLA datasets, with detailed training settings provided in Appendix R.1. As shown in Table 4, integrating Bi-LoRA improves the average score across MRPC and CoLA by 1.45%, achieving gains of up to 1.80% on MRPC and 1.83% on CoLA. These results confirm that Bi-LoRA can be seamlessly integrated with previous approaches and deliver consistent improvements.

Table 4: Results of fine-tuning T5-base on MRPC and CoLA using various LoRA variants, both standalone and combined with Bi-LoRA.

| Method | MRPC | CoLA | Avg. |
|---|---|---|---|
| LoRA-GA | $88.81_{\pm 0.47}$ | $58.87_{\pm 1.00}$ | 73.84 |
| PiSSA | $88.15_{\pm 0.24}$ | $58.66_{\pm 0.47}$ | 73.41 |
| DoRA | $88.81_{\pm 0.29}$ | $59.89_{\pm 0.72}$ | 74.35 |
| LoRA-GA + Bi-LoRA | $89.62_{\pm 0.24}$ | $60.70_{\pm 0.27}$ | 75.16 |
| PiSSA + Bi-LoRA | $\mathbf{89.95}_{\pm 0.50}$ | $59.77_{\pm 0.83}$ | 74.86 |
| DoRA + Bi-LoRA | $89.54_{\pm 0.13}$ | $\mathbf{60.77}_{\pm 0.47}$ | **75.16** |

## 4.6 ABLATIONS AND HYPERPARAMETER SENSITIVITY

**Setting.** In this section, we ablate Bi-LoRA to quantify each component's contribution, and investigate the hyperparameter sensitivity from three factors: (1) auxiliary learning rate $\eta_2$, (2) ranks of primary ($r_1$) and auxiliary $r_2$ LoRA modules, and (3) joint sensitivity to neighborhood radius $\rho$ and auxiliary rank $r_2$. We fine-tune Llama 3.1-8B and evaluate its performance on the instruction following tasks.

**Component-wise ablations of Bi-LoRA.** We examine whether the improvements of Bi-LoRA arise from the additional capacity introduced by the second LoRA branch or from the adversarial ascent step. To this end, we ablate the clipping schemes and the usage of the auxiliary LoRA module.

Table 5 shows that Dual-LoRA, which shares the same two-LoRA architecture as Bi-LoRA but updates both LoRA modules with gradient descent, only slightly outperforms vanilla LoRA (+0.07 Avg.). Intuitively, the parallel descent branch in Dual-LoRA behaves similarly to merging with another LoRA update whose norm is constrained by the global radius $\rho$, which prevents it from fully leveraging the increased rank and representational capacity. This indicates that the performance gains of Bi-LoRA primarily stem from the adversarial ascent step rather than from merely adding an extra LoRA branch. We further compare global clipping with two alternatives: no-clipping and per-layer clipping. Removing clipping leads to noticeable degradation (-3.48 Avg.), as the perturbations become excessively large. And per-layer clipping yields only marginal improvements over LoRA (+0.24 Avg.). These observations support global clipping as the more stable and effective choice. A more detailed explanation of why we adopt global clipping is provided in Appendix G.

**Sensitivity to auxiliary learning rate $\eta_2$.** Table 6 shows that Bi-LoRA is insensitive to auxiliary learning rate $\eta_2$: the average score remains within a narrow band of $[50.55, 51.19]$ as $\eta_2$ varies from 1e-4 to 1e-3. This indicates that Bi-LoRA is robust to $\eta_2$ in a relatively wide range. Interestingly, the best average performance is obtained when $\eta_2 = 3e\text{-}4$, which coincides with the optimal primary learning rate $\eta_1$ given by Ren et al. (2024). This suggests that using a shared learning rate for the primary and auxiliary branches yields more coordinated optimization, consistent with our observation in Table A7. Hence, in all main experiments, we simply set $\eta_2 = \eta_1$, reducing hyperparameter tuning while retaining strong performance.

**Sensitivity to ranks $r_1$ and $r_2$.** Figure 6a shows that the average point rises monotonically up to the auxiliary rank $r_2 = 8$, which we adopt as the default for fine-tuning scenario. Additionally, further increasing $r_2$ does not yield significant improvements or even downgrades the performance.

Table 6: Sensitivity of Bi-LoRA to the auxiliary learning rate $\eta_2$ with Llama 3.1-8B on instruction following tasks, with the same primary learning rate $\eta_1$ as in Section 4.2.

| $\eta_2$ | MMLU | DROP | HEval | BBH | Avg |
|---|---|---|---|---|---|
| 1e-4 | $\mathbf{63.89}_{\pm 0.25}$ | $51.05_{\pm 0.13}$ | $45.12_{\pm 0.35}$ | $42.84_{\pm 0.15}$ | $50.73$ |
| 3e-4 | $63.67_{\pm 0.15}$ | $\mathbf{51.53}_{\pm 0.33}$ | $\mathbf{46.12}_{\pm 0.89}$ | $\mathbf{43.45}_{\pm 0.15}$ | $\mathbf{51.19}$ |
| 5e-4 | $63.47_{\pm 0.35}$ | $51.23_{\pm 0.24}$ | $45.93_{\pm 0.73}$ | $43.05_{\pm 0.12}$ | $50.92$ |
| 1e-3 | $63.40_{\pm 0.17}$ | $51.32_{\pm 0.22}$ | $44.51_{\pm 0.93}$ | $42.96_{\pm 0.23}$ | $50.55$ |

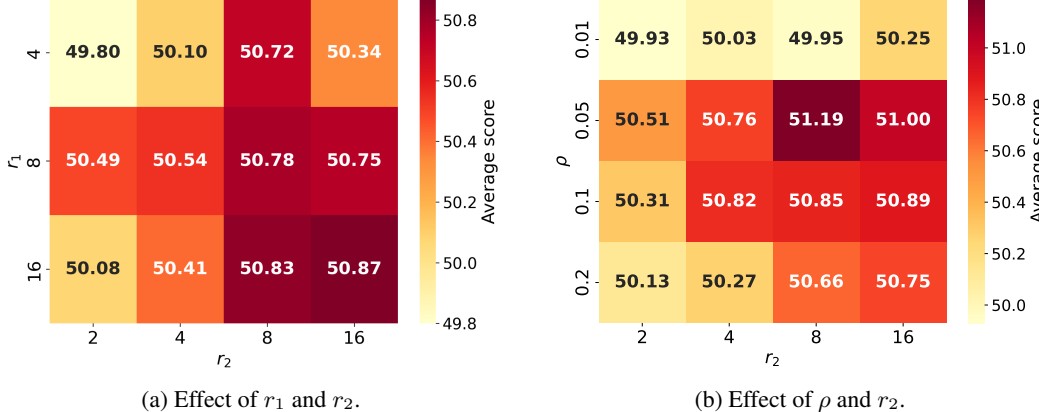

(a) Effect of $r_1$ and $r_2$.        (b) Effect of $\rho$ and $r_2$.

Figure 6: Sensitivity of Bi-LoRA to primary ($r_1$) and auxiliary ($r_2$) LoRA ranks and perturbation radius $\rho$ with Llama 3.1-8B on instruction-following tasks.

**Sensitivity to neighborhood radius $\rho$ and auxiliary rank $r_2$.** Figure 6b further examines the joint effect of $\rho$ and $r_2$. The heatmap also corroborates that enlarging $r_2$ from 2 to 8 is beneficial. Overall, these trends suggest that $\rho$ and $r_2$ govern complementary aspects of neighborhood size and auxiliary capacity and can be tuned largely independently for a given architecture and dataset, in line with previous work of SAM.

### 4.7 TRAINING TIME AND MEMORY COST

Table 7: Peak memory and time per optimization step (relative to LoRA in parentheses).

| Model & Dataset | Method | Memory (GB) | Time (s) |
|---|---|---|---|
| Llama 3.1-8B Cleaned Alpaca | LoRA | 23.69 | 8.93 (100%) |
| | LoRA-SAM | 24.00 | 19.23 (215%) |
| | Bi-LoRA | 24.32 | 9.71 (109%) |

We present detailed comparisons of memory and time per optimization step for LoRA, LoRA-SAM, and Bi-LoRA on Llama 3.1-8B (Cleaned Alpaca). Table 7 shows that Bi-LoRA significantly reduces the LoRA-SAM's training overhead, decreasing it from over 210% to about 110% relative to vanilla LoRA. This gain comes from jointly performing optimization and perturbation, eliminating SAM's extra gradient step. For memory, Bi-LoRA incurs only a minimal overhead ($< 0.7$GB), as it adds a lightweight auxiliary module. Additional comparisons on training time are provided in Appendix S.

## 5 CONCLUSION

In this paper, we propose Bi-LoRA, a novel dual-LoRA variant that leverages an auxiliary LoRA module to enhance the generalization performance of low-rank adaptation. Bi-LoRA decouples optimization from weight perturbation for better optimizing the sharpness of the loss landscape, allowing both to be updated in a single backward pass without requiring additional gradient steps, as in SAM. Extensive experiments across diverse tasks and architectures demonstrate Bi-LoRA's efficiency and effectiveness in improving generalization.

ACKNOWLEDGMENTS

The research leading to these results has received funding from National Key Research Development Project (2023YFF1104202) and National Natural Science Foundation of China (62376155). The authors would like to thank the chairs and reviewers for their thoughtful comments on this paper. Yuhang Liu also gratefully acknowledges the financial support provided by Prof. Xin Yang at Shanghai Jiao Tong University.

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

APPENDIX

## A    ETHICS STATEMENT

This paper proposes Bi-LoRA, a parameter-efficient fine-tuning method that introduces adversarial perturbations to improve generalization of large language models, which can reduce memory and energy costs, and thus broaden access to LLM research.

For dataset use, we use only publicly available datasets under their licenses, with no human subjects or personal data involved.

Regarding safety and privacy, we recognize potential risks. By reducing hardware barriers, Bi-LoRA could also enable malicious actors to efficiently fine-tune models for harmful applications or exacerbate dataset biases if such biases are present. Furthermore, the auxiliary adversarial adapter, while improving robustness, introduces an additional parameter pathway that could hypothetically be exploited for backdoor insertion or undesired behaviors if misused. To mitigate these risks, we will anonymously release only training code and experimental scripts in the supplementary materials that follow community standards for responsible research, and we rely exclusively on open benchmarks with no sensitive or personal data. We encourage future users of Bi-LoRA to adopt similar safeguards, including dataset audits for bias, robust model evaluation, and adherence to ethical guidelines on downstream applications.

## B    REPRODUCIBILITY STATEMENT

We provide detailed experimental settings and hyperparameters in Appendix R to ensure reproducibility. The Bi-LoRA implementation and example experiment scripts will be released anonymously in the **supplementary materials**. All models, benchmarks, and evaluation metrics used in this paper are either open-sourced or explicitly identified when closed-source resources are involved.

## C    LLM USAGE STATEMENT

We use large language models (LLMs) as general-purpose tools at the following stages of this work.

- **Writing.** We employ both closed-source and open-source LLMs solely for text polishing, including grammar correction, clarity improvement, and minor stylistic refinement. Importantly, they are not used to generate research ideas, technical content, and substantive arguments.
- **Benchmarking.** For evaluation on the MT-Bench benchmark, we use the GPT-4 API to obtain GPT-4 scores, following the standard evaluation protocol for MT-Bench.

No part of the research design, methodology, or core scientific contributions relied on LLMs. Their role is restricted to auxiliary assistance in language refinement and standardized benchmark scoring.

## D    DETAILS ON SHARPNESS-AWARE MINIMIZATION (SAM)

Let $\mathcal{L}(W)$ denote the empirical loss of model parameters $W$. Standard ERM can converge to sharp minima where small parameter perturbations significantly increase loss. SAM's goal is to target the training toward flat regions by minimizing the worst-case loss inside a :

$$\min_{W} \max_{\|\varepsilon\| \leq \rho} \mathcal{L}(W + \varepsilon), \tag{A1}$$

where $\rho > 0$ is the neighborhood radius, $\| \cdot \|$ is a kind of norm, and $\varepsilon$ is the perturbation. Intuitively, Eqn, (A1) prefers parameters whose entire neighborhood has uniformly low loss (flat minima), which has been empirically linked to improved generalization. A practical single-step approximation solves the inner maximization by first-order ascent and then updates $W$ using the gradient evaluated at the perturbed point.

Denote by $\mathcal{L}(W; \mathcal{B})$ the mini-batch loss on $\mathcal{B}$ and by $\nabla_W \mathcal{L}$ its gradient. A typical SAM update computes a normalized gradient ascent perturbation and then a descent step on the same mini-batch

---

**Algorithm A1** Sharpness-Aware Minimization (SAM)

---

1: **Input:** Initial weights $W_0$; learning rate $\eta$; radius $\rho$; optimizer $\mathsf{Opt}(\cdot)$; max steps $K$
2: **Output:** Final weights $W$
3: Initialize model weights $W \leftarrow W_0$
4: **for** $t = 0, 1, \ldots, K-1$ **do**
5:     Sample mini-batch $\mathcal{B}_t$
6:     $g \leftarrow \nabla_W \ell(W; \mathcal{B}_t)$                                                   {gradient at clean weights}
7:     $\varepsilon \leftarrow \rho \cdot \dfrac{g}{\|g\|}$                                            {first-order worst-case direction}
8:     $W_{\mathrm{adv}} \leftarrow W + \varepsilon$
9:     $g_{\mathrm{adv}} \leftarrow \nabla_W \mathcal{L}(W_{\mathrm{adv}}; \mathcal{B}_t)$         {use the same batch to produce perturbed gradient}
10:    $W \leftarrow \mathsf{Opt}(W,\ g_{\mathrm{adv}},\ \eta)$                           {e.g., SGD/AdamW step}
11: **end for**
12: **return** $W$

---

$\mathcal{B}_t$, but at different parameter values:

$$\varepsilon_t = \rho \cdot \frac{\nabla_W \mathcal{L}(W_t; \mathcal{B}_t)}{\left\| \nabla_W \mathcal{L}(W_t; \mathcal{B}_t) \right\|}, \tag{A2}$$

$$W_{t+1} \leftarrow W_t \ - \ \eta \, \nabla_W \mathcal{L}(W_t + \varepsilon_t; \mathcal{B}_t), \tag{A3}$$

i.e., two gradient evaluations per iteration. Eqn. (A2) determines the perturbation direction at $W_t$ (clean weight), while Eqn. (A3) computes the update at the perturbed parameter $W_t + \epsilon_t$ (perturbed weight).

## E  RELATED WORK

**Low-Rank Adaptation (LoRA).** LoRA (Hu et al., 2022) is a widely adopted parameter-efficient fine-tuning (PEFT) method. It models weight updates via low-rank matrices without incurring additional inference costs. Many studies have been proposed to improve the performance of LoRA (Koohpayegani et al., 2024; Dou et al., 2024; Tian et al., 2024; Wang et al., 2025; Yen et al., 2025; Wu et al., 2025). On the resource perspective, AdaLoRA (Zhang et al., 2023) dynamically adjusts the rank allocation, while MELoRA (Ren et al., 2024) trains multiple mini-LoRA modules in parallel to cut down on trainable parameters. From the optimization perspective, LoRA-GA (Wang et al., 2024), PISSA (Meng et al., 2024) and MiLoRA (Zhang et al., 2024) accelerate convergence and boost performance through enhanced initialization, and DoRA (Liu et al., 2024) decomposes the adaptation into magnitude and direction for better optimization. In this paper, we enhance LoRA optimization by optimizing the loss landscape sharpness of the full parameter space through an auxiliary LoRA module. Our approach is orthogonal to prior works.

**Sharpness-Aware Minimization (SAM).** SAM formulates the optimization objective as a min-max problem to seek flat minima, encouraging model parameters to reside in regions with consistently low loss values, yielding state-of-the-art generalization and robustness. Despite its effectiveness, SAM doubles the computational cost because its inner maximization is approximated via an additional gradient ascent step, thereby limiting its application to large-scale models. Several SAM variants have been developed to enhance its generalization performance (Kim et al., 2022; Li & Giannakis, 2023; Li et al., 2024c). ASAM (Kwon et al., 2021) introduces adaptive sharpness to address SAM's scale-dependency issue, while GSAM (Zhuang et al., 2022) jointly minimizes surrogate and perturbed losses to locate flatter region. Meanwhile, other works aim to improve SAM's training efficiency (Du et al., 2022b; Jiang et al., 2023; Ji et al., 2024). LookSAM (Liu et al., 2022a) applies adversarial perturbations only periodically. Recent studies have also focused on applying SAM to LoRA fine-tuning, aiming to enhance generalization while maintaining training efficiency. For example, BAR (Li et al., 2024a) makes SAM's implicit balancedness regularization explicit for scale-invariant tasks like LoRA, achieving SAM-level generalization gains with significantly less computation. Flat-LoRA (Li et al., 2025) replaces the costly inner maximization with random weight perturbation to tackle the coupling issue in LoRA-SAM without sacrificing efficiency. In this paper, we introduce a novel framework that integrates SAM with LoRA through dual LoRA modules. Our method employs

auxiliary LoRA modules to generate adversarial perturbations decoupled from optimization, incurring minimal additional overhead.

# F PROOFS

## F.1 PROOF AND DISCUSSION OF PROPOSITION 2: ALIGNMENT OF BI-LORA'S ASCENT DIRECTION WITH PREVIOUS FULL GRADIENT

*Proof.* Let

$$G_t = \nabla_W \mathcal{L}(W_0 + B_{1,t} A_{1,t} + \tilde{\epsilon}_t), \tag{A4}$$

denote the full gradient at step $t$, with $\tilde{\epsilon}_t = B_{2,t} A_{2,t}$ the auxiliary perturbation.

One update step for Bi-LoRA is formalized as

$$\begin{cases} B_{1,t+1} = B_{1,t} - \eta_1 G_t A_{1,t}^\top, & A_{1,t+1} = A_{1,t} - \eta_1 B_{1,t}^\top G_t, \\ B_{2,t+1} = B_{2,t} + \eta_2 G_t A_{2,t}^\top, & A_{2,t+1} = A_{2,t} + \eta_2 B_{2,t}^\top G_t, \end{cases} \tag{A5}$$

where $\eta_1, \eta_2 > 0$ are step sizes.

Expanding the auxiliary perturbation Eqn. (A5) gives

$$\tilde{\epsilon}_{t+1} = B_{2,t+1} A_{2,t+1} = \tilde{\epsilon}_t + \eta_2 \left( G_t A_{2,t}^\top A_{2,t} + B_{2,t} B_{2,t}^\top G_t \right) + \mathcal{O}(\eta_2^2). \tag{A6}$$

Thus, the alignment between the previous gradient Eqn. Eqn. (A4) and the update of the auxiliary perturbation is

$$\langle G_t, \tilde{\epsilon}_{t+1} - \tilde{\epsilon}_t \rangle = \eta_2 \left( \|A_{2,t} G_t^\top\|_F^2 + \|B_{2,t}^\top G_t\|_F^2 \right) + \mathcal{O}(\eta_2^2) \geq 0, \tag{A7}$$

since both terms on the right-hand side are nonnegative (squared Frobenius norms).

Therefore, Eqn. by Eqn. (A7), the auxiliary update $(B_2, A_2)$ always increases the inner objective in the direction of the full gradient Eqn. Eqn. (A4). This proves that the Bi-LoRA perturbation aligns with previous SAM's perturbation direction. □

As reported in several efficient SAM variants, reusing the previous ascent direction can still yield strong generalization (e.g., $S^2$ SAM (Ji et al., 2024) in the sparse training scenario). In general, aligning with the previous, not the current direction, has benefit on efficiency. The reason that aligning with the previous gradient works may be attributed to the low-dimension/low-rank properties of the gradients (Li et al., 2022; Zhao et al., 2024), especially in later stage of training, or in fine-tuning stage. Due to the low-dimension/low-rank property, the consecutive gradient directions tend to exhibit similarity. And the intuition behind Bi-LoRA is that adversarial perturbations across different steps are "continuous": the optimal adversarial perturbation at iteration $t$ is likely to be close to the optimal perturbation at iteration $t + 1$, given that the weight differences between the continuous steps are small under LoRA optimization. This means we can obtain the adversarial perturbation at step $t + 1$ with a "slight" adjustment based on the perturbation at $t$. As a result, this method can effectively optimize the sharpness while maintaining the same training efficiency as regular LoRA training.

## F.2 PROOF AND DISCUSSIONS OF PROPOSITION 3: EQUIVALENCE BETWEEN SAM AND BI-LORA

*Proof.* SAM solves the min–max objective

$$\min_W \max_{\|\epsilon\|_2 \leq \rho} \mathcal{L}(W + \epsilon), \tag{A8}$$

where $\rho > 0$ controls the perturbation radius.

A first-order Taylor expansion gives

$$\mathcal{L}(W + \epsilon) \approx \mathcal{L}(W) + \langle \nabla_W \mathcal{L}(W), \epsilon \rangle. \tag{A9}$$

The inner maximization is

$$\max_{\|\epsilon\|_2 \leq \rho} \langle \nabla_W \mathcal{L}(W), \epsilon \rangle, \tag{A10}$$

which is attained at

$$\epsilon^\star = \rho \frac{\nabla_W \mathcal{L}(W)}{\|\nabla_W \mathcal{L}(W)\|_2}. \tag{A11}$$

Substituting Eqn. (A11) into Eqn. (A8) yields the regularized objective

$$\mathcal{L}(W) + \rho \|\nabla_W \mathcal{L}(W)\|_2. \tag{A12}$$

For Bi-LoRA, the weight matrix is decomposed as $W = W_0 + B_1 A_1 + B_2 A_2$. The training objective is

$$\min_{B_1, A_1} \max_{\substack{B_2, A_2 \\ \mathrm{rank}(B_2 A_2) \leq r, \, \|B_2 A_2\|_F \leq \rho}} \mathcal{L}(W_0 + B_1 A_1 + B_2 A_2), \tag{A13}$$

where the perturbation is explicitly parameterized as

$$\tilde{\epsilon} = B_2 A_2, \quad \mathrm{rank}(\tilde{\epsilon}) \leq r, \quad \|\tilde{\epsilon}\|_F \leq \rho. \tag{A14}$$

At $\theta = W_0 + B_1 A_1$, expanding gives

$$\mathcal{L}(\theta + \tilde{\epsilon}) \approx \mathcal{L}(\theta) + \langle \nabla_\theta \mathcal{L}, \tilde{\epsilon} \rangle. \tag{A15}$$

Thus the inner maximization becomes

$$\max_{\substack{\mathrm{rank}(\tilde{\epsilon}) \leq r \\ \|\tilde{\epsilon}\|_F \leq \rho}} \langle \nabla_\theta \mathcal{L}, \tilde{\epsilon} \rangle. \tag{A16}$$

Let the SVD of $\nabla_\theta \mathcal{L}$ be

$$\nabla_\theta \mathcal{L} = U \Sigma V^\top, \qquad \sigma_1 \geq \sigma_2 \geq \cdots. \tag{A17}$$

The optimum of Eqn. (A16) is achieved at

$$\tilde{\epsilon}^\star = \rho \, U_r V_r^\top, \tag{A18}$$

where $U_r, V_r$ are the top-$r$ singular vectors of $\nabla_\theta \mathcal{L}$, and $\mathrm{rank}(\tilde{\epsilon}^\star) = r$ with $\|\tilde{\epsilon}^\star\|_F = \rho$.

Substituting Eqn. (A18) into Eqn. (A16) gives

$$\langle \nabla_\theta \mathcal{L}, \tilde{\epsilon}^\star \rangle = \rho \sum_{i=1}^r \sigma_i(\nabla_\theta \mathcal{L}) = \rho \|\nabla_\theta \mathcal{L}\|_{(r)}, \tag{A19}$$

where

$$\|\nabla_\theta \mathcal{L}\|_{(r)} := \sum_{i=1}^r \sigma_i(\nabla_\theta \mathcal{L}) \tag{A20}$$

is the Ky Fan $r$-norm.

Hence, Bi-LoRA reduces to the regularized objective

$$\min_{A_1, B_1} \mathcal{L}(W_0 + B_1 A_1) + \rho \|\nabla_{W_0 + B_1 A_1} \mathcal{L}\|_{(r)}. \tag{A21}$$

$$\square$$

When $r$ covers all singular values, the Ky Fan norm becomes the nuclear norm, and Bi-LoRA recovers the SAM objective. Conversely, SAM specializes to Bi-LoRA when only the sharpest $r$ singular directions are penalized.

And Proposition 3 represents the ideal objective of Bi-LoRA and serves to illustrate our motivation intuitively. In practice, however, we adopt a simultaneous optimization approach formalized in Eqn. (9), which does not introduce additional gradient steps and thus maintains efficiency. Experimental results demonstrate that our approximate approach can already significantly improve performance compared to LoRA.

Furthermore, Figure 3 shows that, the auxiliary LoRA module continues broad exploration, while the primary module converges rapidly to a nearly fixed point; once the primary stabilizes, the auxiliary then begins to converge quickly. This behavior suggests that at each parameter point, there exists a (though inefficient) convergent multi-step scheme for the auxiliary module, which is consistent with the intuition of Proposition 3.

Table A1: Results of fine-tuning Llama 3.1-8B on instruction-following tasks with more LoRA-variants and SAM-based methods.

| Method | MMLU | DROP | HEval | BBH | Avg |
|--------|------|------|-------|-----|-----|
| LoRA | $63.38_{\pm 0.39}$ | $49.82_{\pm 0.54}$ | $43.15_{\pm 0.93}$ | $42.82_{\pm 0.27}$ | 49.79 |
| PiSSA | $63.59_{\pm 0.14}$ | $50.20_{\pm 0.20}$ | $43.86_{\pm 0.12}$ | $43.25_{\pm 0.51}$ | 50.22 |
| DoRA | $63.58_{\pm 0.22}$ | $50.53_{\pm 0.10}$ | $44.10_{\pm 0.73}$ | $42.98_{\pm 0.79}$ | 50.30 |
| DeLoRA | $63.49_{\pm 0.32}$ | $51.40_{\pm 0.28}$ | $45.53_{\pm 0.73}$ | $42.72_{\pm 0.29}$ | 50.78 |
| HiRA | $63.62_{\pm 0.33}$ | $51.17_{\pm 0.26}$ | $45.12_{\pm 0.61}$ | $42.89_{\pm 0.20}$ | 50.70 |
| WSAM | $63.42_{\pm 0.12}$ | $50.63_{\pm 0.14}$ | $42.88_{\pm 0.81}$ | $43.16_{\pm 0.52}$ | 50.02 |
| $S^2$-SAM | $\mathbf{63.73}_{\pm 0.38}$ | $50.80_{\pm 0.23}$ | $43.09_{\pm 0.20}$ | $42.44_{\pm 0.19}$ | 50.02 |
| **Bi-LoRA** | $63.67_{\pm 0.15}$ | $\mathbf{51.53}_{\pm 0.33}$ | $\mathbf{46.12}_{\pm 0.89}$ | $\mathbf{43.45}_{\pm 0.15}$ | **51.19** |

## G  REASONS OF USING GLOBAL CLIPPING

We clip using the *total* Frobenius norm over all auxiliary LoRA modules in Bi-LoRA for the following reasons.

- **Faithfulness to SAM.** The SAM-style inner maximization is defined with a *single* $\rho$-ball over the full parameter vector, i.e., $\max_{\|\epsilon\| \leq \rho} \mathcal{L}(w + \epsilon)$. In Bi-LoRA, the perturbation is implemented entirely through the auxiliary LoRA modules, so constraining the *global* Frobenius norm of all auxiliary modules (Eqn. (8)) is the faithful analogue of SAM's original formulation.
- **Avoiding scale explosion and instability from per-layer normalization.** If each layer is normalized independently, every layer receives a perturbation with norm $\rho$, so the overall perturbation magnitude roughly scales with the number of adapters. This not only leads to training instability, but also makes the same $\rho$ incomparable across architectures with different depths or adapter placements. SAM's original global constraint is designed precisely to avoid such architecture-dependent scaling effects.
- **Empirical evidence.** In Table 5, global clipping consistently outperforms both per-layer clipping and no clipping in terms of final performance and training stability under the same $\rho$.

In summary, global clipping over all auxiliary LoRA modules yields a scale-aware neighborhood that is consistent with SAM's objective, comparable across architectures, and empirically more stable, which motivates our choice of this clipping scheme.

## H  MORE BASELINES WITH LLAMA 3.1-8B ON INSTRUCTION FOLLOWING TASKS

Table A1 further compares Bi-LoRA with four LoRA variants (PiSSA, DoRA, HiRA, DeLoRA) and two SAM-based methods (WSAM, $S^2$-SAM) on Llama 3.1-8B instruction following tasks. Bi-LoRA attains the highest average performance of 51.19, outperforming the strongest LoRA variant by about 0.4% and vanilla LoRA by roughly 1.4% points.

## I  RESULTS ON NATURAL LANGUAGE UNDERSTANDING

**Setting.** We fine-tune the T5-base model (Raffel et al., 2020) on multiple datasets from GLUE (Wang et al., 2019b) and SuperGLUE (Wang et al., 2019a) benchmarks, including MNLI, SST2, CoLA, QNLI, MRPC, BoolQ, CB, COPA, RTE, and WIC, following Wang et al. (2024); Li et al. (2025). Performance is evaluated on the development set using accuracy as the primary metric, except for CoLA, where the Matthews correlation coefficient is used.

**Results.** We first focus on the GLUE datasets. From the results in Table A2, we observe that Bi-LoRA outperforms both LoRA and LoRA-SAM, achieving an average improvement of 0.47% and 0.32%, respectively. It is worth noting that Bi-LoRA achieves these gains with the same training speed as

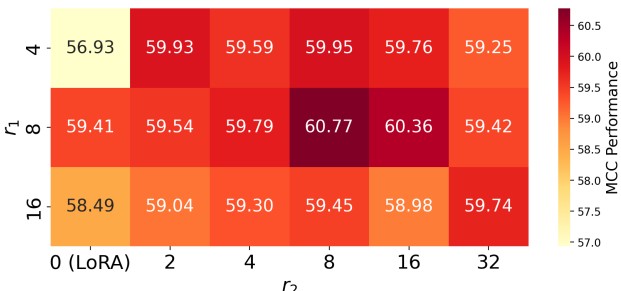

Figure A1: Performance on CoLA under varying ranks of primary ($r_1$) and auxiliary ($r_2$) LoRA modules.

LoRA, whereas LoRA-SAM doubles the training time. Moreover, the gains are more pronounced on smaller datasets, with Bi-LoRA outperforming LoRA by 1.36% on CoLA and 0.82% on MRPC. In contrast, LoRA-SAM shows no clear advantage over vanilla LoRA, indicating its limited ability to enhance generalization due to constrained sharpness optimization.

Next, we evaluate on the SuperGLUE datasets, which feature more challenging language understanding tasks. To reduce variance from the limited evaluation sizes of CB (56), COPA (100), and RTE (277) samples, we increase the number of runs from 3 to 20 for CB and COPA, and to 5 for RTE. Table A3 shows Bi-LoRA demonstrates more significant advantages over LoRA and LoRA-SAM by 0.69% and 0.60% on average, while LoRA-SAM yields minimal gains and even hurts some datasets (e.g., CB). These results confirm the effectiveness of Bi-LoRA in enhancing generalization.

Table A2: Results on fine-tuning the T5-base model on a subset of GLUE datasets. "Cost" indicates the gradient steps per training iteration, e.g., one step (Cost $\times 1$) for Full FT, LoRA, and Bi-LoRA.

| Dataset Size | Cost | MNLI (393k) | SST2 (67k) | CoLA (8.6k) | QNLI (105k) | MRPC (3.7k) | Avg. |
|---|---|---|---|---|---|---|---|
| Full FT | $\times 1$ | $85.57_{\pm 0.09}$ | $94.27_{\pm 0.24}$ | $56.60_{\pm 0.62}$ | $93.18_{\pm 0.09}$ | $87.30_{\pm 0.79}$ | 83.38 |
| LoRA | $\times 1$ | $86.25_{\pm 0.16}$ | $94.23_{\pm 0.30}$ | $59.41_{\pm 0.52}$ | $\mathbf{93.25}_{\pm 0.06}$ | $88.56_{\pm 0.26}$ | 84.34 |
| LoRA-SAM | $\times 2$ | $86.25_{\pm 0.09}$ | $94.46_{\pm 0.17}$ | $59.80_{\pm 0.85}$ | $93.21_{\pm 0.16}$ | $88.73_{\pm 0.52}$ | 84.49 |
| Bi-LoRA | $\times 1$ | $\mathbf{86.33}_{\pm 0.08}$ | $\mathbf{94.34}_{\pm 0.05}$ | $\mathbf{60.77}_{\pm 0.39}$ | $\mathbf{93.25}_{\pm 0.06}$ | $\mathbf{89.38}_{\pm 0.26}$ | $\mathbf{84.81}$ |

Table A3: Results on fine-tuning T5-base with a subset of SuperGLUE datasets. "Cost" indicates the gradient steps per training iteration, e.g., one step (Cost $\times 1$) for Full FT, LoRA, and Bi-LoRA.

| Dataset Size | Cost | BoolQ (9.4k) | CB (0.25k) | COPA (0.4k) | RTE (2.5k) | WIC (5.4k) | Avg. |
|---|---|---|---|---|---|---|---|
| Full FT | $\times 1$ | $72.19_{\pm 0.14}$ | $92.26_{\pm 0.24}$ | $64.67_{\pm 0.26}$ | $84.48_{\pm 0.12}$ | $68.34_{\pm 0.19}$ | 76.39 |
| LoRA | $\times 1$ | $72.20_{\pm 0.21}$ | $\mathbf{92.86}_{\pm 0.27}$ | $63.80_{\pm 0.23}$ | $83.10_{\pm 0.29}$ | $68.60_{\pm 0.32}$ | 76.11 |
| LoRA-SAM | $\times 2$ | $\mathbf{72.47}_{\pm 0.42}$ | $92.32_{\pm 0.28}$ | $64.20_{\pm 0.42}$ | $83.47_{\pm 0.47}$ | $68.55_{\pm 0.76}$ | 76.20 |
| Bi-LoRA | $\times 1$ | $72.25_{\pm 0.30}$ | $\mathbf{92.86}_{\pm 0.32}$ | $\mathbf{64.60}_{\pm 0.25}$ | $\mathbf{83.61}_{\pm 0.37}$ | $\mathbf{70.69}_{\pm 0.40}$ | $\mathbf{76.80}$ |

## J  HYPERPARAMETER SENSITIVITY

### J.1  EFFECT OF THE AUXILIARY RANK ON COLA

To further examine the sensitivity to the auxiliary rank $r_2$, we fine-tune T5-base on the CoLA dataset with $r_1 \in \{4, 8, 16\}$ and $r_2 \in \{2, 4, 8, 16, 32, 64\}$.

Figure A1 shows the performance of Bi-LoRA with varying the primary ($r_1$) and auxiliary ($r_2$) LoRA ranks. We observe that integrating the auxiliary module consistently enhances performance. Notably, $r_2 = 8$ generally delivers good results, which replicates the pattern we reported in Section 4.6, confirming that the sweet-spot generalizes from NLU tasks to instruction-tuning. Therefore, we suggest using $r_2 = 8$ as a task-agnostic default.

## J.2 Effect of the Neighborhood Radius $\rho$

From Table A4, Bi-LoRA performs consistently well across $\rho = \{0.01, 0.05, 0.1\}$, showing its effectiveness to moderate perturbations during training. The ability to achieve strong performance across a range of $\rho$ indicates that Bi-LoRA can effectively adapt to varying levels of perturbation, demonstrating its practicality for diverse applications.

Table A4: Results on CoLA, SST2 and GSM8K under different values of the neighborhood radius $\rho$.

| $\rho$ | CoLA | SST2 | GSM8K |
|---|---|---|---|
| 0.01 | $60.15_{\pm 0.36}$ | $60.15_{\pm 0.36}$ | 59.36 |
| 0.05 | $\mathbf{60.77}_{\pm 0.39}$ | $60.15_{\pm 0.36}$ | 59.74 |
| 0.1 | $60.18_{\pm 0.59}$ | $60.15_{\pm 0.36}$ | $\mathbf{60.65}$ |
| 0.2 | $58.95_{\pm 0.81}$ | $60.15_{\pm 0.36}$ | 58.83 |

## K Qualitative Results on SDXL

As discussed in Section 4.4, Bi-LoRA improves both image–text (I2T) and text–text (T2T) CLIP similarity on the 3D Icons dataset. To visualize these gains, Figure A2 presents generated images from SDXL models fine-tuned by LoRA and Bi-LoRA under identical promptss.

**Observations.** Across instances, Bi-LoRA tends to preserve the intended icon identity and style more faithfully. These qualitative differences align with the measured CLIP improvements, i.e., larger average CLIP I2T/T2T scores shown in Table 3 also exhibit clearer textual correspondence in Figure A2.

## L Convergence Analysis of Main and Auxiliary LoRA Modules

We measure the cosine similarity between the weights of the main and auxiliary LoRA modules and their inal weights on CoLA and SST2 with T5-base. The results indicate that the auxiliary module converges substantially slower than the main one, preserving flexibility for sharpness-aware updates.

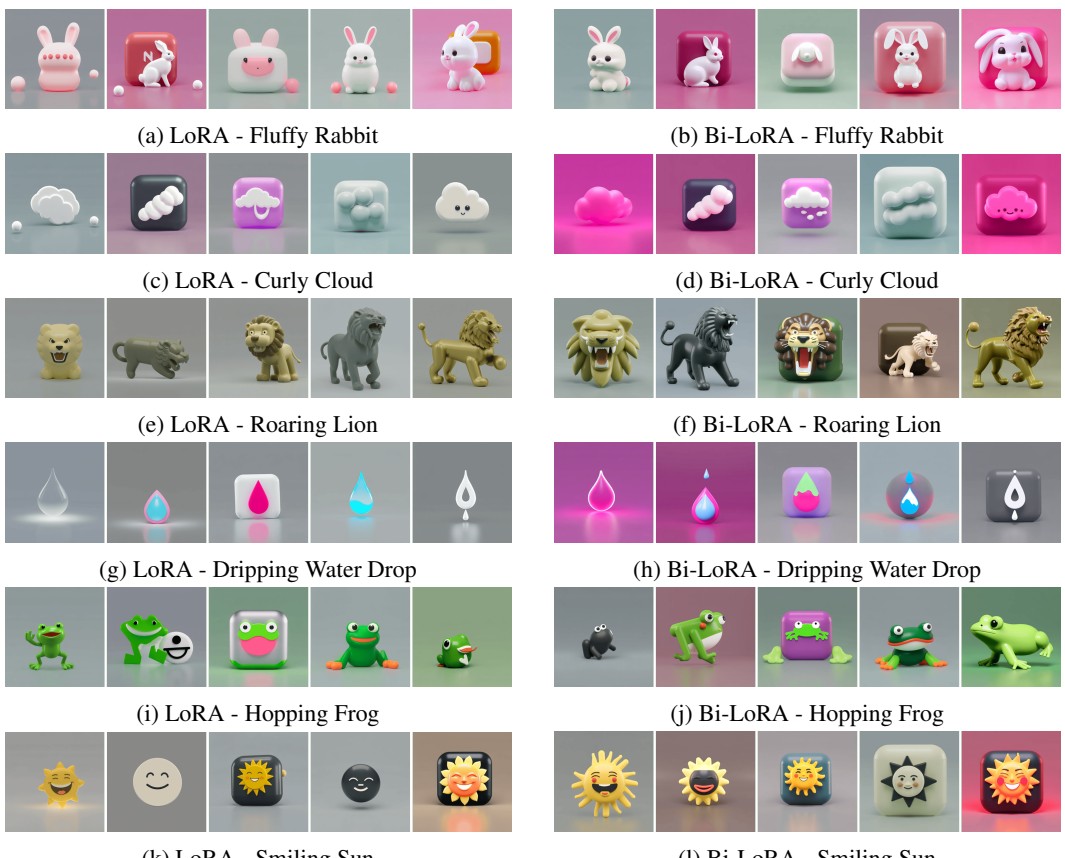

(a) LoRA - Fluffy Rabbit

(b) Bi-LoRA - Fluffy Rabbit

(c) LoRA - Curly Cloud

(d) Bi-LoRA - Curly Cloud

(e) LoRA - Roaring Lion

(f) Bi-LoRA - Roaring Lion

(g) LoRA - Dripping Water Drop

(h) Bi-LoRA - Dripping Water Drop

(i) LoRA - Hopping Frog

(j) Bi-LoRA - Hopping Frog

(k) LoRA - Smiling Sun

(l) Bi-LoRA - Smiling Sun

Figure A2: Images generated with SDXL fine-tuned using LoRA (left) and Bi-LoRA (right) on the 3D icon dataset. Each row corresponds to a different prompt in the form "*a ToK icon of a <Instance>, in the style of ToK*". Images in the same row are generated with the same random seed for fair comparison.

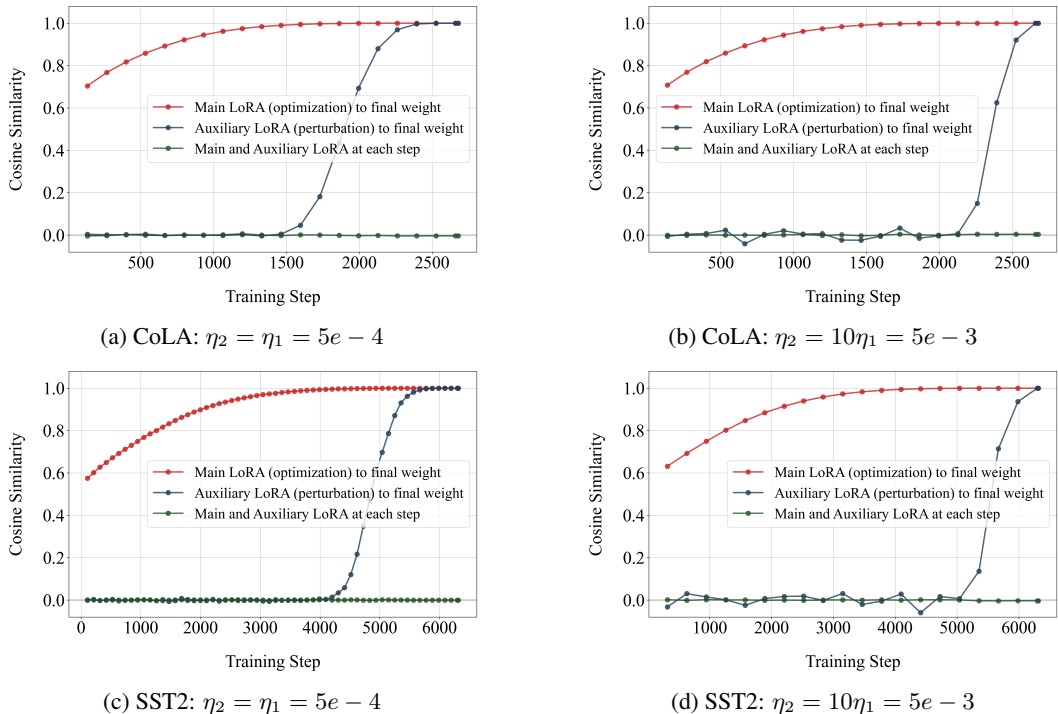

Figure A3: Cosine similarity between the main and auxiliary LoRA weights and each final weights during training on CoLA and SST2 using T5-base, under different auxiliary learning rates ($\eta_2$). **The auxiliary module (perturbation, blue lines) converges in the final 20% of steps, significantly slower than the main (optimization, red lines) module**. A larger $\eta_2 = 10\eta_1$ further slows down convergence. Moreover, the auxiliary module's trajectory remains independent of the main module throughout training (green lines).

## M    BI-LORA IN QUANTIZED SETTINGS

We apply Bi-LoRA in a quantized setting, where the transition to lower precision can cause the model to jump from a flat region into a sharper region with higher local loss, reducing generalization on unseen data. Following QLoRA (Dettmers et al., 2023), we quantize the pretrained model to NF4 and adopt the setup described in Section 5.3.

As shown in Table A5, Bi-QLoRA outperforms QLoRA and QLoRA-SAM by 2.58% and 2.36% on GSM8K, respectively. This gain is attributed to Bi-LoRA's ability to maintain flatness under quantization by decoupling optimization and perturbation during training.

Table A5: Performance on GSM8K under quantized NF4 settings. Results are averaged over 3 runs.

| Method | GSM8K (%) |
|---|---|
| QLoRA | $57.77_{\pm 0.39}$ |
| QLoRA-SAM | $57.89_{\pm 0.29}$ |
| Bi-QLoRA | $\mathbf{59.26}_{\pm 0.17}$ |

## N    EFFECT ON ADVERSARIAL ROBUSTNESS

Figure 4 already shows that Bi-LoRA is more robust than vanilla LoRA against random perturbations applied both in the LoRA subspace and in the full parameter space. We now investigate robustness to adversarial perturbations. Concretely, we fine-tune T5-base on CoLA and report Matthew's

Table A6: Results on CoLA with adversarial parameter perturbation. Perturbation strength $\alpha$ from 0 to 5.0.

| | Perturb in LoRA space | | | | | |
| --- | --- | --- | --- | --- | --- | --- |
| Method | $\alpha = 0$ | $\alpha = 0.5$ | $\alpha = 1.0$ | $\alpha = 2.5$ | $\alpha = 4.0$ | $\alpha = 5.0$ |
| LoRA | 60.62 | 59.42 | 57.47 | 42.07 | 24.22 | 19.19 |
| LoRA-SAM | 60.95 | 59.14 | 58.44 | 52.12 | 41.33 | 30.61 |
| Bi-LoRA | 61.12 | 59.36 | 59.14 | 48.47 | 33.24 | 24.20 |
| | Perturb in full space | | | | | |
| Method | $\alpha = 0$ | $\alpha = 0.5$ | $\alpha = 1.0$ | $\alpha = 2.5$ | $\alpha = 4.0$ | $\alpha = 5.0$ |
| LoRA | 60.62 | 52.69 | 44.42 | 19.08 | 11.39 | 6.55 |
| LoRA-SAM | 60.95 | 56.16 | 45.67 | 24.77 | 18.02 | 13.34 |
| Bi-LoRA | 61.12 | 56.29 | 49.00 | 32.91 | 24.83 | 18.06 |

Correlation Coefficient. Following (Kim et al., 2022), we apply the worst-case parameter perturbation

$$\theta \to \theta + \alpha \frac{\nabla_\theta \mathcal{L}}{\|\nabla_\theta \mathcal{L}\|}, \tag{A22}$$

evaluating attacks restricted to (i) the LoRA subspace ($\theta \in \{B, A\}$) and (ii) the full parameter space ($\theta = W$), with perturbation strength $\alpha$ from 0.5 to 5.0. Results are shown in Table A6.

In the LoRA subspace, Bi-LoRA consistently suffers smaller performance degradation than vanilla LoRA. LoRA-SAM shows the strongest resilience, which aligns with its explicit optimization of sharpness in this subspace. In the full parameter space, degradation is more substantial across all methods, yet Bi-LoRA still mitigates the impact more effectively than LoRA-SAM.

These findings are consistent with our random perturbation results, demonstrating that Bi-LoRA provides robustness to adversarial parameter perturbations, particularly when the attack targets the full parameter space.

## O  TRAINING AND TEST METRIC CURVES

We fine-tune T5-base on MRPC for 10 epochs, tracking both training and eval losses and metrics. The results in A4 shows nearly identical training losses across methods, yet Bi-LoRA attains a lower validation loss and higher accuracy. Importantly, the smaller train–val generalization gap demonstrates Bi-LoRA's improved generalization.

## P  COMPUTE RESOURCES

We utilize two types of GPUs: the NVIDIA RTX 4090 24GB and the A100 80GB. For NLU and Text-to-Image generation tasks, computations are performed on a single RTX 4090. All other experiments are conducted on a single A100.

## Q  LEARNING RATES FOR TUNING THE TWO MODULES IN BI-LORA

In our experiments, both modules in Bi-LoRA use the same learning rate of $5 \times 10^{-4}$. Initially, we explored an unequal learning rate strategy. However, results in Table A7 show that tuning $\eta_2$ provides only a limited improvement in performance. Considering the trade-off between gains and the additional effort for hyperparameter tuning, we recommend using the same learning rate for both modules.

## R  EMPIRICAL DETAILS

This section provides detailed empirical setups used for the previous experiments.

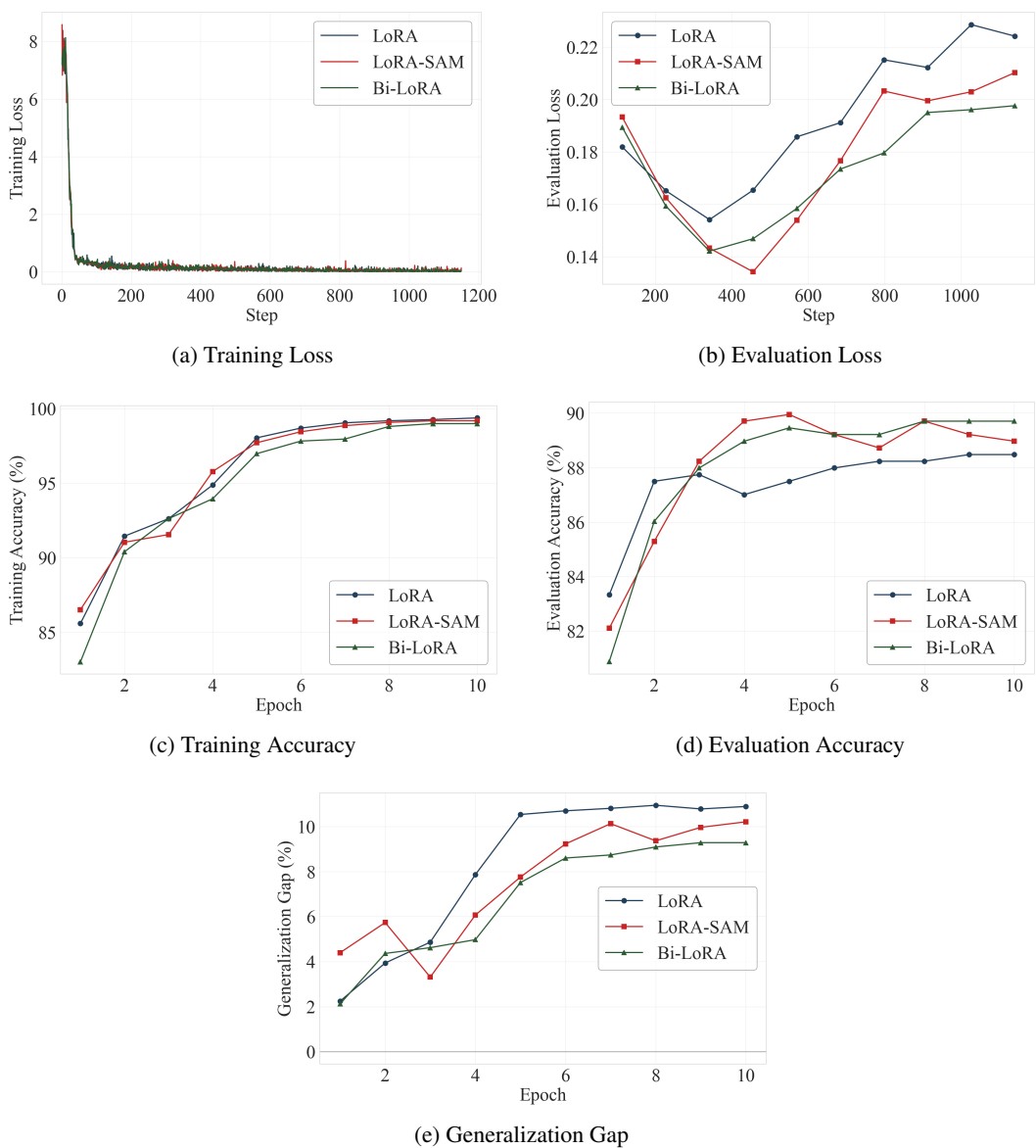

Figure A4: Training and evaluation loss, accuracy, and generalization gap of T5-base fine-tuned on the MRPC dataset.

Table A7: CoLA accuracy under different learning rate of the auxiliary module $\eta_2$ settings (with $\eta_1 = 5 \times 10^{-4}$ for the main LoRA module).

| $\eta_2$ | $1 \times 10^{-4}$ | $5 \times 10^{-4}$ (main) | $5 \times 10^{-3}$ |
|---|---|---|---|
| CoLA | $60.23_{\pm 0.35}$ | $60.77_{\pm 0.55}$ | $60.44_{\pm 0.87}$ |

We tune the neighborhood radius $\rho$ over $\{0.005, 0.01, 0.05, 0.1, 0.2, 0.5\}$ for LoRA-SAM and Bi-LoRA and search learning rates among $\{2e-5, 5e-5, 1e-4, 2e-4\}$ for Full FT and $\{5e-5, 1e-4, 2e-4, 5e-4, 1e-3\}$ for LoRA and its variants.

## R.1 EXPERIMENTS ON THE GLUE AND SUPERGLUE DATASETS

The hyperparameter configurations for all GLUE and SuperGLUE experiments (Sections I, 4.5 and 4.6) are detailed in Tables A8 and A9. As discussed in Section I, we set the neighborhood radius $\rho$ to 0.01 and 0.05 for LoRA-SAM and Bi-LoRA, respectively, in the GLUE experiments. For SuperGLUE, we use $\rho$ values of 0.05 and 0.1 for LoRA-SAM and Bi-LoRA, respectively. For full fine-tuning, we set the learning rate to 1e-4 while keeping all other hyperparameters unchanged. We use the same learning rate of 5e-4 for both the primary and auxiliary LoRA modules in Bi-LoRA.

Table A8: Hyperparameter configurations for fine-tuning T5-base using LoRA-based methods on the GLUE datasets.

| Hyperparameter | MNLI | SST-2 | COLA | QNLI | MRPC |
|---|---|---|---|---|---|
| Learning Rate | | | 5e-4 | | |
| Batch Size | | | 32 | | |
| Epochs | 3 | 3 | 10 | 3 | 10 |
| Max Sequence Length | | | 256 | | |
| LoRA Rank | | | 8 | | |
| LoRA Alpha | | | 16 | | |
| LR Scheduler | | | Cosine | | |
| Target Modules | | | All | | |
| Warmup Ratio | | | 0.03 | | |
| Evaluation Metric | Accuracy | Accuracy | Matthews Corr. | Accuracy | Accuracy |

Table A9: Hyperparameter settings for fine-tuning T5-base using LoRA-based methods on the SuperGLUE datasets.

| Hyperparameter | BoolQ | CB | COPA | RTE | WIC |
|---|---|---|---|---|---|
| Learning Rate | | | 5e-4 | | |
| Batch Size | | | 32 | | |
| Epochs | 10 | 50 | 50 | 10 | 10 |
| Max Sequence Length | | | 256 | | |
| LoRA Rank | | | 8 | | |
| LoRA Alpha | | | 16 | | |
| LR Scheduler | | | Cosine | | |
| Target Modules | | | All | | |
| Warmup Ratio | | | 0.03 | | |
| Evaluation Metric | | | Accuracy | | |

## R.2 EXPERIMENTS ON LLAMA MODELS

The hyperparameter settings for Llama 2-7B and Llama 3.1-8B are listed in Tables A10 and A11.

The neighborhood radius $\rho$ is set to 0.01 and 0.05 for LoRA-SAM and Bi-LoRA. For full fine-tuning, we set the micro-batch size to 4, 2, 2, and 8 for the respective four tasks, and the learning rate is set to 5e-5. We use the same learning rate of 5e-4 for both the primary and auxiliary LoRA modules in Bi-LoRA.

Table A10: Hyperparameter configurations for fine-tuning Llama 2-7B on Math, Code, and Chat tasks using LoRA-based methods. "Math", "Code" and "Chat" correspond to mathematical reasoning, code generation, and dialogue generation tasks, respectively, as described in Section 4.2.

| Hyperparameter | Math | Code | Chat |
|---|---|---|---|
| Learning rate | | 5e-4 | |
| Batch Size | | 32 | |
| Micro-Batch size | | 4 | |
| Epochs | | 2 | |
| Max Sequence Length | | 1024 | |
| LoRA Rank | | 8 | |
| LoRA Dropout | | 0.0 | |
| LoRA Alpha | | 16 | |
| Target Modules | | All | |
| LR Scheduler | | Cosine | |
| Warmup Ratio | | 0.03 | |

Table A11: Hyperparameter configurations for fine-tuning Llama 3.1-8B on Cleaned Alpaca dataset.

| Hyperparameter | Cleaned Alpaca |
|---|---|
| Learning rate | 3e-4 |
| Batch Size | 128 |
| Micro-Batch size | 4 |
| Epochs | 3 |
| Max Sequence Length | 256 |
| LoRA Rank | 8 |
| LoRA Dropout | 0.0 |
| LoRA Alpha | 16 |
| Target Modules | All |
| LR Scheduler | Cosine |
| Warmup Ratio | 0.03 |

## R.3 EXPERIMENTS ON QWEN MODEL

We first grid-search the learning rate $\in \{1e-5, 5e-5, 1e-4, 3e-4, 5e-4\}$ for LoRA, Bi-LoRA and LoRA-SAM, and perturbation radius $\rho \in \{0.1, 0.05, 0.01\}$. For all random noise baselines we tune the variance $\rho^2 \in \{0.1, 0.01, 0.005, 0.001\}$.

We set the perturbation radius to $\rho = 0.1$ for Bi-LoRA and $\rho = 0.05$ for LoRA-SAM and other SAM variants, use variance $\rho^2 = 0.001$ for all random-noise baselines. Other hyperparameters are summarized in Table A12.

Table A12: Hyperparameter configurations for fine-tuning Qwen 2.5-14B on Cleaned Alpaca dataset.

| Hyperparameter | Cleaned Alpaca |
| --- | --- |
| Learning rate | 5e-5 |
| Batch Size | 128 |
| Micro-Batch size | 4 |
| Epochs | 3 |
| Max Sequence Length | 256 |
| LoRA Rank | 8 |
| LoRA Dropout | 0.0 |
| LoRA Alpha | 16 |
| Target Modules | All |
| LR Scheduler | Cosine |
| Warmup Ratio | 0.03 |

## R.4 EXPERIMENTS ON DIFFUSION MODELS

The finetuning dataset, 3D Icons[1], contains 23 training images, all featuring a square icon.

The hyperparameter settings for Diffusion Models are listed in Table A13. We finetune the model for 500 steps with a constant learning rate of 2e-4, and set the $\rho$ for Bi-LoRA to 0.05. The LoRA rank is set to 4, with an auxiliary rank of 4 for Bi-LoRA. We use the same learning rate of 2e-4 for both the primary and auxiliary LoRA modules in Bi-LoRA. The instance prompt used for DreamBooth training is "a TOK icon, in the style of TOK", and the validation prompt used for generating the specific icons is formatted as "a ToK icon of a <Instance>, in the style of ToK". We follow the scripts implemented by Hugging Face[2].

Table A13: Hyperparameter settings for fine-tuning SDXL using Dreambooth on the 3D Icons dataset

| Hyperparameter | Value |
| --- | --- |
| Learning Rate | 2e-4 |
| Batch Size | 4 |
| Micro-Batch Size | 1 |
| Train Steps | 500 |
| Resolution | 1024 |
| Validation Epochs | 25 |
| LoRA Rank $r$ | 4 |
| LoRA Dropout | 0.0 |
| LoRA Alpha $\alpha$ | 8 |
| Target Modules | $W_K, W_Q, W_V, W_O$ |
| LR Scheduler | Constant |
| Warmup Steps | 0 |

---

[1] https://huggingface.co/datasets/linoyts/3d_icon
[2] https://github.com/huggingface/diffusers/blob/main/examples/dreambooth/README_sdxl.md

## R.5 Efficient SAM Variants for LoRA Fine-Tuning

To ensure a fair comparison, we adopt the results of Flat-LoRA from Li et al. (2025), and follow the default setup of Li et al. (2024a) for LoRA-oBAR/nBAR with $\alpha = 0.25$. And following Du et al. (2022a), (Liu et al., 2022b), LoRA-ESAM perturbs 50% of the parameters on the top-50% sharpness-sensitive data, while LoRA-LookSAM applies SAM perturbation every five steps. All other hyperparameters are aligned with those used in our prior experiments.

## S Time Comparisons Across Tasks

In this section, we compare the per-step optimization time for LoRA, LoRA-SAM, and Bi-LoRA across a range of benchmarks. The results are organized based on different datasets and tasks, including T5 experiments on the GLUE and SuperGLUE datasets, as well as Llama experiments on domain-specific datasets. All experiments were conducted on a single NVIDIA RTX 4090 GPU 24GB, using the same hyperparameters as in the previous experiments.

Table A14: Optimization time (s) per step for T5-base on GLUE benchmarks. The average time across tasks is also reported.

| Dataset | MNLI | SST2 | CoLA | QNLI | MRPC | Avg. Time |
|---------|------|------|------|------|------|-----------|
| Full FT | 0.758 | 0.538 | 0.549 | 0.756 | 0.681 | 0.654 |
| LoRA | 0.458 | 0.310 | 0.273 | 0.508 | 0.442 | 0.398 |
| LoRA-SAM | 1.036 | 0.573 | 0.546 | 0.950 | 1.039 | 0.829 |
| Bi-LoRA | 0.476 | 0.325 | 0.305 | 0.513 | 0.508 | 0.425 |

Table A15: Optimization time (s) per step for T5-base on SuperGLUE benchmarks. The average time across tasks is also reported.

| Dataset | BoolQ | CB | COPA | RTE | WIC | Avg. Time |
|---------|-------|-----|------|-----|-----|-----------|
| Full FT | 0.622 | 0.415 | 0.451 | 0.689 | 0.544 | 0.544 |
| LoRA | 0.276 | 0.276 | 0.253 | 0.263 | 0.286 | 0.271 |
| LoRA-SAM | 0.519 | 0.534 | 0.568 | 0.688 | 0.484 | 0.559 |
| Bi-LoRA | 0.304 | 0.285 | 0.278 | 0.276 | 0.262 | 0.281 |

Table A16: Optimization time (s) per step for training Llama2 on math (MetaMathQA), code (Code-Feedback) and chat (WizardLM) tasks.

| Method | MetaMathQA | Code-Feedback | WizardLM |
|--------|------------|---------------|----------|
| LoRA | 2.81 | 5.10 | 5.13 |
| LoRA-SAM | 5.52 | 10.10 | 10.53 |
| ROP | 2.88 | 5.41 | 5.24 |
| RWP_full | 3.21 | 6.02 | 5.88 |
| RWP_LoRA | 2.97 | 5.56 | 5.38 |
| LoRA-oBAR | 2.90 | 5.41 | 5.21 |
| LoRA-nBAR | 2.89 | 5.41 | 5.24 |
| Flat-LoRA | 3.42 | 5.92 | 5.78 |
| LoRA-ESAM | 4.20 | 8.13 | 7.35 |
| LoRA-LookSAM | 3.47 | 6.49 | 6.29 |
| Bi-LoRA | 3.12 | 5.68 | 5.43 |

Table A17: Optimization time (s) per step for training Llama 3-8B on the instruction-following dataset Cleaned Alpaca.

| Method | Cleaned Alpaca |
|---|---|
| LoRA | 8.93 |
| LoRA-SAM | 19.23 |
| ROP | 9.43 |
| RWP_full | 12.35 |
| RWP_LoRA | 10.10 |
| LoRA-oBAR | 9.35 |
| LoRA-nBAR | 9.35 |
| Flat-LoRA | 9.90 |
| LoRA-ESAM | 13.44 |
| LoRA-LookSAM | 11.49 |
| Bi-LoRA | 9.71 |

Table A18: Optimization time (s) per step for training Qwen 2.5-14B on the instruction-following dataset Cleaned Alpaca.

| Method | Cleaned Alpaca |
|---|---|
| LoRA | 15.47 |
| LoRA-SAM | 32.37 |
| ROP | 15.87 |
| RWP_full | 21.28 |
| RWP_LoRA | 16.67 |
| LoRA-oBAR | 15.87 |
| LoRA-nBAR | 15.87 |
| Flat-LoRA | 17.06 |
| LoRA-ESAM | 21.79 |
| LoRA-LookSAM | 19.07 |
| Bi-LoRA | 16.63 |

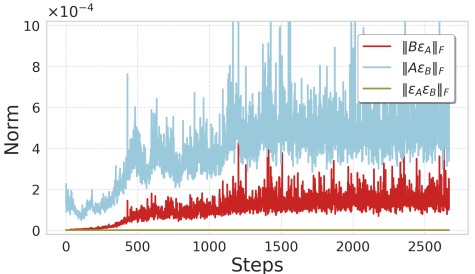

(a) Norm of different components in the key of the first self-attention layer in the fifth encoder block

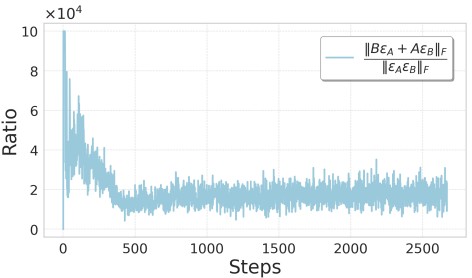

(b) Norm ratio of the key of the first self-attention layer in the fifth encoder block

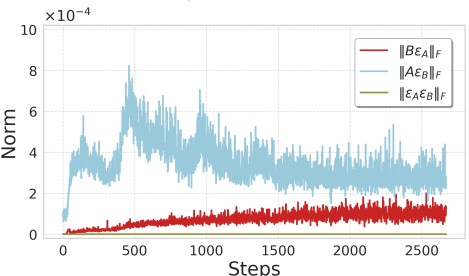

(c) Norm of different components in the output projection the second self-attention layer in the eighth encoder block

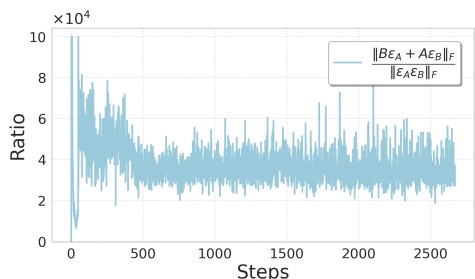

(d) Norm ratio of the output projection in the second self-attention layer in the eighth encoder block

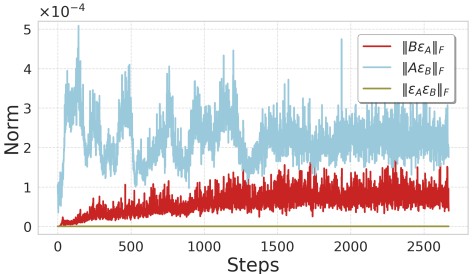

(e) Norm of different components in the value of the first self-attention layer in the ninth encoder block

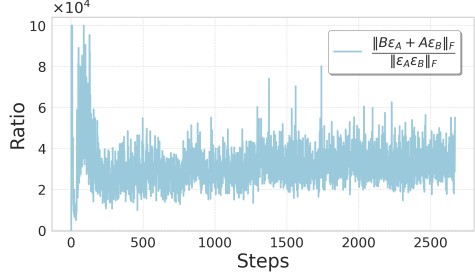

(f) Norm ratio of the value of the first self-attention layer in the ninth encoder block

Figure A5: Analyses of training statistics for LoRA-SAM. (a) and (c) and (e): Frobenius norms of different terms. And (b), (d) and (f): ratio of the Frobenius norms of the first two terms ($B\epsilon_A + \epsilon_B A$) to that of the third term ($\epsilon_B\epsilon_A$) in Eqn. (5) during fine-tuning. Models are fine-tuned on CoLA with T5 for 10 epochs.

## T    NORM AND RATIO ANALYSIS ACROSS LAYERS

In addition to Figures 2a and 2b, we extend the analysis of LoRA-SAM's training dynamics to additional self-attention layers across different encoder blocks to verify the consistency of our observations.

Figures A5a, A5c and A5e show the Frobenius norms of different terms in the key, output projection and value of self-attention layers in the 5th, 8th, and 9th encoder blocks, respectively. And Figures A5b, A5d and A5f present corresponding norm ratios. As observed, the norm of the third higher-order term remains several orders of magnitude smaller than the first two terms, confirming its negligible impact.

## U  RESTRICTING PERTURBATION IN THE OPTIMIZATION SUBSPACE LIMITS LORA-SAM

We evaluate the performance of LoRA-SAM on Llama 3.1-8B for the instruction following benchmark, sweeping the rank over $r \in \{4, 8, 16, 32, 64\}$. Table A19 shows that while the average score rises monotonically with rank, the gain per doubling quickly reduces. The upward trend confirms that restricting perturbation in the optimization subspace does limit LoRA-SAM, and that increasing the rank partially mitigates it. Ideally, if $r = \max\{m, n\}$ for each weight matrix $W \in \mathbb{R}^{m \times n}$, LoRA-SAM becomes full parameter SAM, and the limitation would disappear. Our results show that even at $r = 64$, LoRA-SAM still lags behind our Bi-LoRA with primary rank = 8, auxiliary rank = 8, whose wider exploratory subspace is achieved with decoupling the (auxiliary) perturbation and (primary) optimization subspace. Moreover, LoRASAM's SAM-style perturbation doubles computation, whereas Bi-LoRA keeps the cost close to vanilla LoRA.

Table A19: Instruction-tuning results on fine-tuning Llama 3.1-8B using LoRA-SAM with varying rank.

| Rank | MMLU | DROP | HEval | BBH | Avg. |
|------|------|------|-------|-----|------|
| 4 | 64.16 | 48.66 | 40.24 | 43.37 | 49.11 |
| 8 | 65.50 | 50.93 | 40.85 | 43.14 | 50.11 |
| 16 | 66.05 | 51.92 | 40.85 | 42.91 | 50.43 |
| 32 | 64.45 | 50.93 | 44.51 | 42.43 | 50.58 |
| 64 | 65.81 | 50.72 | 43.90 | 42.37 | 50.70 |

## V  THEORETICAL COMPARISON WITH OTHER EFFICIENT SAM VARIANTS DESIGNED FOR LORA FINE-TUNING

We provide a brief theoretical comparison showing the difference between Bi-LoRA and two efficient SAM variants, LoRA-o/nBAR and Flat-LoRA.

Starting from the common SAM objective $\min_W \max_{\|\epsilon\| \leq \rho} \mathcal{L}(W + \epsilon)$, all three methods approximate this minimax objective under a low-rank constraint, yet they differ in how the approximation is carried out. LoRAo/nBAR utitlizes SAM's implicit regularization, it replace the inner maximization with an explicit balanceness penalty, leading to $\min_{B_1, A_1} \mathcal{L}(W_0 + BA) + \rho \|B^\top B - AA^\top\|$ (take oBAR and $\|B\| \geq \|A\|$ for example) with a looser $O(\rho^2)$ approximation. Flat-LoRA and Bi-LoRA both target LoRA-SAM's restricted-subspace issue, i.e., the perturbation $\epsilon \approx BB^\top (\nabla_W \mathcal{L}) + (\nabla_W \mathcal{L}) A^\top A$ is confined to $\mathrm{Col}(B)$ and $\mathrm{Row}(A)$. Yet they pursue flatness differently. Flat-LoRA replaces the inner maximisation with a random weight perturbation (RWP) expectation, optimising $\min_{B_1, A_1} \mathbb{E}_{\epsilon \sim \mathcal{N}(0, \sigma^2)} \mathcal{L}(W_0 + B_1 A_1 + \epsilon)$, thus encouraging expected flatness rather than worst-case sharpness. Bi-LoRA introduces a gradient-driven auxiliary adapter ( $B_2, A_2$ ) so that $\min_{B_1, A_1} \max_{\|B_2 A_2\| \leq \rho} \mathcal{L}(W_0 + B_1 A_1 + B_2 A_2)$, allowing a more targeted exploration of the sharp regions through learnable perturbations. From the perspective of gradient-norm regularization, Bi-LoRA essentially replaces SAM's $\ell_2$-norm constraint with a rank $r$ nuclear norm constraint, thereby inheriting similar theoretical guarantees on generalization and convergence.

