# OpenReview forum: "Bi-LoRA: Efficient Sharpness-Aware Minimization for Fine-Tuning Large-Scale Models"
_ICLR.cc/2026/Conference — ICLR 2026 Poster_

### Official Review · Reviewer_K3nD · 2025-10-16

**Soundness:** 2
**Presentation:** 3
**Contribution:** 2
**Rating:** 4
**Confidence:** 4

**Summary:**

To address the generalization issue of LoRA, this paper proposes a method called Bi-LoRA. Specifically, besides the standard LoRA module trained via gradient descent, the method introduces an auxiliary LoRA module trained via gradient ascent. By training these two modules in parallel, the approach achieves an effect analogous to sharpness-aware minimization (SAM). Extensive experiments demonstrate that the proposed method attains superior or comparable performance across various settings.

**Strengths:**

1. The proposed method is simple and easy to implement, while effectively reducing computational overhead compared to traditional SAM.

2. The experiments are comprehensive, covering both image generation and language tasks.

3. The paper is clearly written and easy to follow.

**Weaknesses:**

1. The motivation of the paper is confusing. It is unclear why the computation of perturbations needs to consider a broader space. Moreover, the proposed method does not seem to alleviate this issue, as the perturbation space it explores is still constrained.
2. Although the proposed method is theoretically shown in Proposition 3 to be equivalent to SAM under the corresponding assumptions, this equivalence does not hold in practice due to the conflict between its assumptions and the observed experimental behavior.
3. Similarly, the argument presented in Proposition 2 is not convincing, as the update direction of the auxiliary module at iteration t is applied to the optimization of iteration t+1, making its alignment with the SAM perturbation direction at iteration t conceptually questionable.
4. The improvement brought by the proposed method is quite limited; however, this is understandable.

**Questions:**

1. In Figure 3, the convergence curve of the auxiliary module appears to converge rapidly after the primary module has already converged. Providing a reasonable explanation for this phenomenon would help deepen the understanding of the role and effect of the auxiliary module.

---

> ### Author Response · Authors · 2025-11-21
> **Response to Reviewer K3nD (part 1)**
>
> ### R4.1. Why perturbations for LoRA finetuning need larger exploration, and broader exploration is achieved by Bi-LoRA.
>
> > W1: The motivation of the paper is confusing. It is unclear why the computation of perturbations needs to consider a broader space. Moreover, the proposed method does not seem to alleviate this issue, as the perturbation space it explores is still constrained.
>
> Thank you for your question. Our main motivation is computational efficiency: approximate SAM-style perturbation for generalization with only LoRA's single-pass, low-memory training.
>
> The reason for considering a broader perturbation space is that, during inference, LoRA modules are merged into the pretrained weights. Therefore, it is necessary to optimize the sharpness of the full parameter space. However, LoRA‑SAM’s adversarial updates are low‑rank, and its perturbation and optimization spaces coincide. Figure 2c shows that the LoRA subspace converges early in training; once this collapse occurs, the inner maximization is confined to the optimization subspace and can no longer probe high‑curvature directions that are orthogonal to it.
>
> Your observation that Bi-LoRA’s perturbation space is still constrained is correct. Though Bi-LoRA's perturbation space Col(B2) is still a subspace, the key difference from LoRA-SAM is that Bi-LoRA decouples the low-rank weight perturbation from the primary LoRA modules $(B_1, A_1)$. In Figure 3, we show that before the primary module reaches a plateau, the auxiliary adapter continues broad exploration with no sign of convergence, even though it is low-rank, which enables better tracking of sharpness during training while preserving memory efficiency. In contrast, the weight perturbation in LoRA-SAM is heavily determined by $(B_1, A_1)$.
>
> ### R4.2. Ideal theoretical result in Proposition 3.
>
> > W2: Although the proposed method is theoretically shown in Proposition 3 to be equivalent to SAM under the corresponding assumptions, this equivalence does not hold in practice due to the conflict between its assumptions and the observed experimental behavior.
>
> Thank you for raising this point. As you mentioned, Proposition 3 represents the ideal objective of Bi-LoRA and serves to intuitively illustrate our motivation. In practice, however, we adopt a simultaneous optimization approach formalized in Eqn. (9), which does not introduce additional gradient steps and thus maintains efficiency. Experimental results demonstrate that our approximate approach can already significantly improve performance compared to LoRA.
>
> Furthermore, Figure 3 shows that, the auxiliary LoRA module continues broad exploration, while the primary module converges rapidly to a nearly fixed point; once the primary stabilizes, the auxiliary then begins to converge quickly. This behavior suggests that at each parameter point, there exists a (though inefficient) convergent multi-step scheme for the auxiliary module, which is consistent with the intuition of Proposition 3.
>
> ### R4.3. Why does the alignment with the previous perturbation in Proposition 2 help.
>
> > W3: Similarly, the argument presented in Proposition 2 is not convincing, as the update direction of the auxiliary module at iteration t is applied to the optimization of iteration t+1, making its alignment with the SAM perturbation direction at iteration t conceptually questionable.
>
> Your intuition is quite insightful. Proposition 2 indeed shows that the auxiliary update is aligned with the previous step's gradient, not the current one. Less ideal but still works: using the last step’s ascent direction can still yield strong generalization. This has been reported in several efficient SAM variants (e.g., $S^2$ SAM [1] in the sparse training scenario) that reuse the previous ascent direction. In general, aligning with the previous, not the current direction, has benefit on efficiency. The reason that aligning with the previous gradient works may attribute to the low-dimension/low-rank properties of the gradients [2, 3], especially in later stage of training, or in fine-tuning stage. Due to the low-dimension/low-rank property, the consecutive gradient directions tend to exhibit similarity.
>
> [1] A Single-Step, Sharpness-Aware Minimization is All You Need to Achieve Efficient and Accurate Sparse Training
>
> [2] Low Dimensional Landscape Hypothesis is True: DNNs can be Trained in Tiny Subspaces
>
> [3] GaLore: Memory-Efficient LLM Training by Gradient Low-Rank Projection
>
> ### R4.4. Limited improvement of Bi-LoRA.
>
> > W4: The improvement brought by the proposed method is quite limited; however, this is understandable.
>
> Thank you for your understanding. The possible reason is that we are working with a high baseline, for example, by tuning the optimal learning rates for different methods to ensure a fair comparison and then evaluate the true effectiveness of our approach. As a result, there may be limited room for further improvement.

---

> > ### Author Response · Authors · 2025-11-21
> > **Response to Reviewer K3nD (part 2)**
> >
> > ### R4.5. Explaining the late-stage rapid convergence of the auxiliary LoRA module.
> >
> > > Q1: In Figure 3, the convergence curve of the auxiliary module appears to converge rapidly after the primary module has already converged. Providing a reasonable explanation for this phenomenon would help deepen the understanding of the role and effect of the auxiliary module.
> >
> > Thanks for raising such an insightful and interesting question. The role of the auxiliary module is to “actively” search for the worst-case weight perturbation in order to optimize the sharpness of the entire parameter space. As the primary module converges, the sharpness optimization performed by the auxiliary module tends to focus on a single weight point within a norm-$\rho$ ball, which leads to the subsequent convergence of the auxiliary LoRA module. It is important to note that the convergence of the auxiliary module is generally much slower than that of the main module (the main converges in the first ~20% steps, while the auxiliary in the final ~20% steps). This behavior indicates that Bi-LoRA intentionally preserves a longer exploration phase for the perturbation subspace, enabling broader sharpness search before settling.

---

### Official Review · Reviewer_6yuC · 2025-10-27

**Soundness:** 3
**Presentation:** 4
**Contribution:** 3
**Rating:** 8
**Confidence:** 5

**Summary:**

This paper proposes Bi-LoRA, introducing a *bi-directional* (dual-module) structure for integrating Sharpness-Aware Minimization (SAM) into Low-Rank Adaptation (LoRA). The method adds an auxiliary LoRA module that performs gradient ascent to model adversarial perturbations, while the primary LoRA module performs gradient descent for task adaptation.

This design allows both updates to be computed simultaneously in one forward/backward pass, avoiding the doubled training cost of standard SAM, while also decoupling sharpness optimization from task adaptation.

Empirical results on a broad range of tasks (LLMs like LLaMA-2/3.1, Qwen-14B, and SDXL diffusion models) show that Bi-LoRA improves generalization performance over LoRA and LoRA-SAM with only minor additional memory and time cost.

**Strengths:**

1. **Clear Motivation.** The paper clearly identifies the inefficiency and subspace limitation of LoRA-SAM and logically motivates the need for decoupling SAM’s sharpness optimization from LoRA’s adaptation.
2. **Elegant Design.** The proposed bi-directional scheme (gradient descent for main LoRA, ascent for auxiliary LoRA) is simple yet conceptually neat. It transforms SAM’s two sequential steps into a single parallel step without additional training time.
3. **Theoretical Analysis.** The paper provides analytical support (Propositions 1–3), clarifying LoRA-SAM’s subspace restriction and proving that Bi-LoRA’s ascent direction aligns with SAM’s perturbation direction.
4. **Strong Empirical Validation.** Comprehensive experiments across architectures (transformers, diffusion models) and tasks (math reasoning, code, chat, instruction following, and image generation) show consistent improvement. The results are credible and diverse.
5. **Efficiency Retained.** The method achieves near-LoRA efficiency (≈1.09× training time, +0.7 GB memory) while roughly halving SAM’s cost, making it practical for large-scale LLM fine-tuning.
6. **Compatibility with Existing Methods.** The paper demonstrates that Bi-LoRA can be seamlessly integrated with LoRA variants such as LoRA-GA, PiSSA, and DoRA, yielding further gains.

**Weaknesses:**

1. **Overclaim**.  Bi-LoRA has a higher loss in Figure 4(a) than the LoRA-SAM, thus, it is a overclaim that Bi-LoRA achieves a significantly flatter loss landscape in LoRA parameter space in Line 252-253.
2. **Ablation Clarity**. The effect of each component (e.g., gradient clipping in Eqn. (10)) is not fully isolated. It’s unclear how much each contributes to final performance.

**Questions:**

1. How does the choice of the neighborhood radius (ρ) and the auxiliary LoRA rank (r_2)? Is there a scaling rule or heuristic for choosing these jointly?
2. Why you apply the normalization only when the total Frobenius norm over $N$ auxiliary LoRA modules instead of do normalization for each auxiliary LoRA modules?

---

> ### Author Response · Authors · 2025-11-21
> **Response to Reviewer 6yuC (part 1)**
>
> ### R3.1. Overclaim.
>
> > W1: Overclaim. Bi-LoRA has a higher loss in Figure 4(a) than the LoRA-SAM, thus, it is a overclaim that Bi-LoRA achieves a significantly flatter loss landscape in LoRA parameter space in Line 252-253.
>
> Thanks for pointing out this issue. We will revise the claim as "And Figure 4 confirms that Bi-LoRA achieves a significantly flatter loss landscape in the full parameter space, leading to improved generalization".
>
> ### R3.2. Ablations | each component’s contribution of Bi-LoRA.
>
> > W2: Ablation Clarity. The effect of each component (e.g., gradient clipping in Eqn. (10)) is not fully isolated. It’s unclear how much each contributes to final performance.
>
> Thanks for the suggestion. To quantify each component’s contribution, we add the following ablations, and the results are shown in Table R3.2.
> 1. Clipping: per-layer clipping and no clipping (contrasted with our default global clipping).
> 2. Adversarial mechanism: Dual-LoRA, where both LoRA modules are optimized by gradient descent under the same ranks and clipping. At inference, we keep only the primary modules for Dual-LoRA, as in Bi-LoRA.
> 3. Auxiliary module: vanilla LoRA (single module).
>
> We find that Dual-LoRA only slightly improves vanilla LoRA, whereas Bi-LoRA with adversarial ascent achieves a notably higher average score of 51.19, with clear gains especially on DROP (+1.71 over LoRA and +1.56 over Dual-LoRA) and HumanEval (+2.97 and +3.41, respectively). This suggests that simply introducing a second LoRA module and training it jointly by descent does not improve generalization. And the improvement stems from the adversarial ascent dynamics, not from extra representational capacity. Moreover, the clipping ablation in Table R2.3-c further shows that the effect is driven by our global clipping scheme (R3.4 further explains why Bi-LoRA needs global clipping). We again thank you for suggesting these ablations, which help clarify where Bi-LoRA’s gains truly come from.
>
> Table R3.2: Instruction-tuning results on Llama 3.1-8B: ablations on clipping, adversarial ascent, and auxiliary module.
> |Method|Clipping|Adversarial|MMLU|DROP|HEval|BBH|Avg|
> |-|-|-|-|-|-|-|-|
> |LoRA (single module)|✗|✗|63.38$_{\pm 0.39}$|49.82$_{\pm 0.54}$|43.15$_{\pm 0.93}$|42.82$_{\pm 0.27}$|49.79|
> |Dual-LoRA (both descent)|global|✗|63.49$_{\pm 0.12}$|49.97$_{\pm 0.17}$|42.71$_{\pm 0.47}$|43.25$_{\pm 0.15}$|49.86|
> |Bi-LoRA (no clipping)|✗|✓|61.33$_{\pm 0.42}$|47.31$_{\pm 3.66}$|40.24$_{\pm 0.93}$|41.96$_{\pm 0.48}$|47.71|
> |Bi-LoRA (per-layer clip)|per-layer|✓|63.49$_{\pm 0.26}$|50.26$_{\pm 0.30}$|43.29$_{\pm 0.61}$|43.06$_{\pm 0.28}$|50.03|
> |**Bi-LoRA (global clip)**|global|✓|**63.67$_{\pm 0.15}$**|**51.53$_{\pm 0.33}$**|**46.12$_{\pm 0.89}$**|**43.45$_{\pm 0.15}$**|**51.19**|

---

> > ### Author Response · Authors · 2025-11-21
> > **Response to Reviewer 6yuC (part 2)**
> >
> > ### R3.3. Ablations | tuning perturbation radius rho and auxiliary rank r2.
> >
> > > Q1: How does the choice of the neighborhood radius (ρ) and the auxiliary LoRA rank (r_2)? Is there a scaling rule or heuristic for choosing these jointly?
> >
> > Thank you for raising this. In the main text, we observe a broad optimum at $r_2=8$ on Llama 3.1-8B instruction-tuning (Figure 6 in Section 4.6) and the same pattern on T5 with CoLA (Appendix H1). Therefore, we recommend $r_2 = 8$ as a task-agnostic default for fine-tuning.
> >
> > For the perturbation radius $\rho$, we evaulate $\rho\in$ {0.01, 0.05, 0.1, 0.2} across multiple tasks, and in our experiments $\rho=0.05$ or $0.1$ is typically optimal or sub-optimal, which aligns with common recommendations in SAM variants.
> >
> > We also conduct a joint search over $r_2$ and $\rho$, with $r_2 \in$ {2, 4, 8, 16} and $\rho \in$ {0.01, 0.05, 0.10, 0.20}, the results are reported in Table R3.3. For each fixed $r_2$, increasing $\rho$ from $0.01$ to $0.05$–$0.10$ consistently improves the average score, while a larger radius $\rho = 0.20$ leads to a degradation. For each fixed $\rho$, enlarging $r_2$ from 2 to 8 yields clear gains, and $r_2=16$ brings only marginal improvements, which is consistent with our analysis in Section 4.6. Generally, given the same architecture and dataset, $\rho$ is independent of the new hyperparameter $r_2$, in line with previous work on SAM.
> >
> > Table R3.3: Instruction-tuning results on fine-tuning Llama 3.1-8B with Bi-LoRA, varying perturbation radius $\rho$ and auxiliary rank $r_2$.
> > |(r2,rho)|MMLU|DROP|HEval|BBH|Avg|
> > |-|-|-|-|-|-|
> > |(2,0.01)|63.39$_{\pm 0.21}$|50.44$_{\pm 0.21}$|43.29$_{\pm 0.61}$|42.58$_{\pm 0.35}$|49.93|
> > |(2,0.05)|63.40$_{\pm 0.17}$|**51.06$_{\pm 0.09}$**|**44.30$_{\pm 1.02}$**|**43.29$_{\pm 0.23}$**|**50.51**|
> > |(2,0.10)|63.38$_{\pm 0.09}$|50.45$_{\pm 0.28}$|44.10$_{\pm 0.54}$|**43.29$_{\pm 0.15}$**|50.31|
> > |(2,0.20)|**63.53$_{\pm 0.12}$**|50.04$_{\pm 0.17}$|43.90$_{\pm 1.47}$|43.03$_{\pm 0.30}$|50.13|
> >
> > |(r2,rho)|MMLU|DROP|HEval|BBH|Avg|
> > |-|-|-|-|-|-|
> > |(4,0.01)|**63.74$_{\pm 0.14}$**|50.04$_{\pm 0.21}$|43.49$_{\pm 1.13}$|42.85$_{\pm 0.24}$|50.03|
> > |(4,0.05)|63.53$_{\pm 0.12}$|51.05$_{\pm 0.20}$|44.92$_{\pm 0.81}$|**43.55$_{\pm 0.13}$**|50.76|
> > |(4,0.10)|63.46$_{\pm 0.17}$|**51.21$_{\pm 0.17}$**|**45.32$_{\pm 0.54}$**|43.29$_{\pm 0.23}$|**50.82**|
> > |(4,0.20)|63.62$_{\pm 0.19}$|50.34$_{\pm 0.14}$|43.70$_{\pm 0.61}$|43.42$_{\pm 0.29}$|50.27|
> >
> > |(r2,rho)|MMLU|DROP|HEval|BBH|Avg|
> > |-|-|-|-|-|-|
> > |(8,0.01)|63.47$_{\pm 0.25}$|50.16$_{\pm 0.20}$|43.09$_{\pm 0.41}$|43.08$_{\pm 0.30}$|49.95|
> > |(8,0.05)|63.67$_{\pm 0.15}$|**51.53$_{\pm 0.33}$**|**46.12$_{\pm 0.89}$**|43.45$_{\pm 0.15}$|**51.19**|
> > |(8,0.10)|63.53$_{\pm 0.11}$|50.64$_{\pm 0.12}$|45.32$_{\pm 0.81}$|**43.89$_{\pm 0.06}$**|50.85|
> > |(8,0.20)|**63.76$_{\pm 0.30}$**|50.34$_{\pm 0.23}$|44.92$_{\pm 1.08}$|43.61$_{\pm 0.18}$|50.66|
> >
> > |(r2,rho)|MMLU|DROP|HEval|BBH|Avg|
> > |-|-|-|-|-|-|
> > |(16,0.01)|63.76$_{\pm 0.14}$|49.90$_{\pm 0.18}$|44.10$_{\pm 1.02}$|43.23$_{\pm 0.18}$|50.25|
> > |(16,0.05)|**63.86$_{\pm 0.12}$**|**51.35$_{\pm 0.21}$**|45.32$_{\pm 0.54}$|43.48$_{\pm 0.15}$|**51.00**|
> > |(16,0.10)|63.73$_{\pm 0.11}$|50.92$_{\pm 0.20}$|45.12$_{\pm 0.81}$|**43.79$_{\pm 0.14}$**|50.89|
> > |(16,0.20)|63.71$_{\pm 0.14}$|50.45$_{\pm 0.21}$|**45.53$_{\pm 1.47}$**|43.31$_{\pm 0.15}$|50.75|
> >
> > ### R3.4. The reason of using global normalization.
> >
> > > Q2: Why you apply the normalization only when the total Frobenius norm over auxiliary LoRA modules instead of do normalization for each auxiliary LoRA modules?
> >
> > Thank you for your insightful question. We normalize using the total Frobenius norm over all auxiliary LoRA modules for the following reasons:
> >
> > 1. **Faithfulness to SAM**. The SAM-style inner maximization applies a single $\rho$-ball constraint over the full parameter, i.e., $\max _{\|\epsilon\| \leq \rho} \mathcal{L}(w+\epsilon)$. In Bi-LoRA, the perturbation is produced solely by the auxiliary modules, so normalization on the total auxiliary LoRA norm (Eq. (8)) is the correct analogue of SAM's goal.
> > 2. **Inconsistent tuning and training instability of per-layer normalization.** If each layer is normalized separately, this assigns a norm-$\rho$ perturbation to every layer, making the overall perturbation scale grow with the number of adapters, leading to training instability. The same value of $\rho$ would become incomparable across architectures, resulting in inconsistent tuning. SAM’s formulation is built upon such global clipping precisely to avoid the scale dependencies.
> > 3. **Empirical support**: In our reply to reviewer bc19 in R2.3 (Table R2.3-c), we additionally compared clipping forms and found that global clipping outperforms per-layer clipping and no clipping in both final performance and stability under the same $\rho$.
> >
> > Above all, the global normalization generates a scale-aware neighborhood, yielding more reliable optimization. And that's why we do normalization over all auxiliary modules.

---

### Official Review · Reviewer_bc19 · 2025-10-30

**Soundness:** 3
**Presentation:** 3
**Contribution:** 4
**Rating:** 8
**Confidence:** 3

**Summary:**

The paper introduces Bi-LoRA, a dual-adapter fine-tuning scheme in which a primary LoRA module is optimized by gradient descent for task adaptation while an auxiliary LoRA module is optimized by gradient ascent to approximate SAM-style perturbations in a single forward/backward pass. The decoupled design aims to mitigate the perturbation-subspace restriction observed when applying SAM directly to LoRA (LoRA-SAM) and to promote flatter minima in the full parameter space with minimal extra memory and near-LoRA time cost. The authors provide analytical support (e.g., alignment of the auxiliary ascent with the full gradient and a connection to a low-rank gradient-norm/Ky Fan regularizer under ideal inner maximization) and report consistent empirical improvements over LoRA and efficient SAM variants across LLaMA-2/3.1, Qwen-2.5-14B, T5-base (GLUE/SuperGLUE), and SDXL DreamBooth. At inference, the auxiliary adapter is discarded and only the primary LoRA is merged into the base model.

**Strengths:**

1) The method is simple and practical: a bi-directional LoRA scheme that achieves SAM-like sharpness-aware training in a single forward/backward pass with near-LoRA compute and modest memory overhead.
2) The paper offers clear analytical insights, including a derivation of LoRA-SAM’s perturbation subspace limitation and supporting theory for the auxiliary ascent direction (e.g., gradient alignment and a Ky Fan–style regularization view).
3) Empirical results are broad and consistent across LLMs and diffusion models, showing gains over LoRA and efficient SAM baselines, compatibility with other LoRA variants, and robustness under quantization.

**Weaknesses:**

1) The causal source of gains is under-isolated: without a “dual-LoRA (both descent)” control, it remains uncertain how much improvement comes from adversarial ascent versus added optimization degrees of freedom during training.
2) Theoretical grounding for the practical single-step inner ascent is limited; results rely on an idealized inner maximization, leaving a gap between the theory and the implemented algorithm.
3) Ablations and baselines are incomplete: the choice of global vs. per-layer clipping, sensitivity to ρ/rank/step sizes, and comparisons to broader single-pass SAM approximations / LoRA variants are not fully explored.

**Questions:**

1) Can you report a dual-LoRA (both descent) control with the same rank and clipping as Bi-LoRA, to isolate the effect of adversarial ascent?
2) How sensitive are results to ρ, (r1, r2), and the ascent/descent step sizes, and is there a regime where multiple inner ascent steps improve performance at acceptable compute cost?
3) Did you evaluate per-layer clipping (vs. global) and can you provide compute-parity details (hardware, precision, gradient checkpointing) to ensure fair efficiency comparisons across baselines?

---

> ### Author Response · Authors · 2025-11-21
> **Response to Reviewer bc19 (part 1)**
>
> ### R2.1. Ablations | isolating the effect of adversarial ascent.
>
> > W1: The causal source of gains is under-isolated: without a “dual-LoRA (both descent)” control, it remains uncertain how much improvement comes from adversarial ascent versus added optimization degrees of freedom during training.
>
> > Q1: Can you report a dual-LoRA (both descent) control with the same rank and clipping as Bi-LoRA, to isolate the effect of adversarial ascent?
>
> Your suggestion is quite interesting. Following that, we now add an extra experiment: Dual-LoRA (both modules updated by descent) with the same ranks ($r_1=r_2=8$) and global clipping as Bi-LoRA, fine-tuned with Llama 3.1-8B on the instruction following tasks. At inference, we keep only the primary modules for both methods. Results are reported in Table R2.1.
>
> We find that Dual-LoRA only slightly outperforms vanilla LoRA ($r=8$). This is because the additional parallel descent branch in Dual-LoRA can be roughly regarded as as merging with a LoRA module whose norm is bounded by $\rho$, which limits its ability to fully exploit the increased rank and corresponding representational capacity. And this suggests that the performance gains of Bi-LoRA mainly stem from the adversarial ascent step rather than simply introducing an extra LoRA branch. In addition, the results in Table R2.3-c further rule out the influence of the specific clipping form. Again, we sincerely thank you for the suggestion on Dual-LoRA, which helped us better clarify that the generalization improvement primarily comes from the adversarial perturbation.
>
> Table R2.1: Instruction-tuning results on fine-tuning Llama 3.1-8B with LoRA, Dual-LoRA and Bi-LoRA.
> | Method                    | MMLU                   | DROP                   | HEval                  | BBH                    | Avg       |
> | ------------------------- | ---------------------- | ---------------------- | ---------------------- | ---------------------- | --------- |
> | LoRA                      | 63.38$_{\pm 0.39}$     | 49.82$_{\pm 0.54}$     | 43.15$_{\pm 0.93}$     | 42.82$_{\pm 0.27}$     | 49.79     |
> | Dual-LoRA (both descent)  | 63.49$_{\pm 0.12}$     | 49.97$_{\pm 0.17}$     | 42.71$_{\pm 0.47}$     | 43.25$_{\pm 0.15}$     | 49.86     |
> | **Bi-LoRA** (global clip) | **63.67$_{\pm 0.15}$** | **51.53$_{\pm 0.33}$** | **46.12$_{\pm 0.89}$** | **43.45$_{\pm 0.15}$** | **51.19** |
>
> ### R2.2. Gap between the theory and the implementation.
>
> > W2: Theoretical grounding for the practical single-step inner ascent is limited; results rely on an idealized inner maximization, leaving a gap between the theory and the implemented algorithm.
>
> > Q2: is there a regime where multiple inner ascent steps improve performance at acceptable compute cost
>
> Thank you for your concerns on the theoretical grounding. Our contribution focus on efficiency: Bi-LoRA uses a single gradient step, avoiding LoRA-SAM's doubled cost. And this creates a gap between the theory and the implemented algorithm. A nontrivial generalization theory is difficult and left for future work.
>
> In the spirit of LookSAM’s periodic activation of SAM perturbations [1], we could adapt it to Bi-LoRA to periodically perform multi-step inner ascent to obtain a more accurate perturbation direction. And this could further improve performance. However, performing k inner steps on the auxiliary module is impractical: it adds k extra gradient steps and increases compute. Under our current implementation, this trade-off is unfavorable, so we leave such schedules as an interesting direction for future algorithmic improvements.
>
> [1] Towards Efficient and Scalable Sharpness-Aware Minimization.

---

> ### Author Response · Authors · 2025-11-21
> **Response to Reviewer bc19 (part 2)**
>
> ### R2.3. Ablations and more baselines.
> > W3: Ablations and baselines are incomplete: the choice of global vs. per-layer clipping, sensitivity to ρ/rank/step sizes, and comparisons to broader single-pass SAM approximations / LoRA variants are not fully explored.
>
> > Q2: How sensitive are results to ρ, (r1, r2), and the ascent/descent step sizes
>
> > Q3: Did you evaluate per-layer clipping (vs. global) and can you provide compute-parity details (hardware, precision, gradient checkpointing) to ensure fair efficiency comparisons across baselines?
>
> In response to your suggestions, we provide sensitivity analyses for the perturbation radius $\rho$ (Table R2.3-a) and the learning rate of the auxiliary modules lr2 (Table R2.3-b). For lr1 of the descent branch, Bi-LoRA reduces to standard LoRA, and to ensure a strong baseline we first tune the primary learning rate lr2 or follow the protocol of proir work (e.g., MeLoRA [1]), fixing the best $\text{lr}_1$ thereafter.
>
> Sensitivity to ranks $r_1, r_2$ has  been analyzed in Section 4.6: increasing $r_1$ generally helps, and $r_1$ already offers sufficient capacity for our fine-tuning tasks; raising $r_2$ from 2 to 8 consistently improves performance, and further increases yield marginal gains, so we recommend $r_2=8$.
>
> And we conduct ablation for clipping. Beyond the original global clipping, we ablate no-clipping and per-layer clipping and report their impact in Table R2.3-c. And in R3.4, we justify our choice: SAM’s inner maximization applies a single global $\rho$-ball over all parameters; since Bi-LoRA’s perturbation comes solely from the auxiliary adapters, enforcing a total-norm budget is the faithful analogue (i.e., match the SAM constraint). In contrast, per-layer clipping assigns a norm-$\rho$ perturbation for each layer, making the total perturbation incomparable across architectures and inflating the perturbation budget as adapters increase, empirically raising instability in our early experiments, so we keep global clipping as default.
>
> Also, we include two recent LoRA variants, DeLoRA [2] and HiRA [3], and a single-pass SAM variant, $S^2$ SAM [4] as additional baselines (Table R2.3-d).
>
> Across these experiments, we use a single A100 GPU with no gradient checkpointing. Following LoRA-GA [5], we use bf16 precision for the backbone, and fp32 for LoRA parameters.
>
> Table R2.3-a: Sensitivity of Bi-LoRA to perturbation radius $\rho$ on Llama 3.1-8B instruction tuning.
> |$\rho$|MMLU|DROP|HEval|BBH|Avg|
> |-|-|-|-|-|-|
> |0.01|63.47$_{\pm 0.25}$|50.16$_{\pm 0.20}$|43.09$_{\pm 0.41}$|43.08$_{\pm 0.30}$|49.95|
> |0.05|63.67$_{\pm 0.15}$|**51.53$_{\pm 0.33}$**|**46.12$_{\pm 0.89}$**|43.45$_{\pm 0.15}$|**51.19**|
> |0.1|63.53$_{\pm 0.11}$|50.64$_{\pm 0.12}$|45.32$_{\pm 0.81}$|**43.89$_{\pm 0.06}$**|50.85|
> |0.2|**63.76$_{\pm 0.30}$**|50.34$_{\pm 0.23}$|44.92$_{\pm 1.08}$|43.61$_{\pm 0.18}$|50.66|
>
> Table R2.3-b: Sensitivity of Bi-LoRA to auxiliary modules' learning rate lr2 on Llama 3.1-8B instruction tuning.
> |(lr1,lr2)|MMLU|DROP|HEval|BBH|Avg|
> |-|-|-|-|-|-|
> |(3e-4,1e-4)|**63.89$_{\pm 0.25}$**|51.05$_{\pm 0.13}$|45.12$_{\pm 0.35}$|42.84$_{\pm 0.15}$|50.73|
> |(3e-4,3e-4)|63.67$_{\pm 0.15}$|**51.53$_{\pm 0.33}$**|**46.12$_{\pm 0.89}$**|**43.45$_{\pm 0.15}$**|**51.19**|
> |(3e-4,5e-4)|63.47$_{\pm 0.35}$|51.23$_{\pm 0.24}$|45.93$_{\pm 0.73}$|43.05$_{\pm 0.12}$|50.92|
> |(3e-4,1e-3)|63.40$_{\pm 0.17}$|51.32$_{\pm 0.22}$|44.51$_{\pm 0.93}$|42.96$_{\pm 0.23}$|50.55|
>
> Table R2.3-c: Effect of clipping strategy in Bi-LoRA on Llama 3.1-8B instruction tuning.
> |Method|MMLU|DROP|HEval|BBH|Avg|
> |-|-|-|-|-|-|
> |Bi-LoRA (no clipping)|61.33$_{\pm 0.42}$|47.31$_{\pm 3.66}$|40.44$_{\pm 1.08}$|41.96$_{\pm 0.48}$|47.76|
> |Bi-LoRA (per-layer clipping)|63.49$_{\pm 0.26}$|50.26$_{\pm 0.30}$|43.29$_{\pm 0.61}$|43.06$_{\pm 0.28}$|50.03|
> |**Bi-LoRA** (global clipping)|**63.67$_{\pm 0.15}$**|**51.53$_{\pm 0.33}$**|**46.12$_{\pm 0.89}$**|**43.45$_{\pm 0.15}$**|**51.19**|
>
> Table R2.3-d: Comparison with additional single-pass SAM and LoRA variants on Llama 3.1-8B instruction tuning.
> |Method|MMLU|DROP|HEval|BBH|Avg|
> |-|-|-|-|-|-|
> |LoRA|63.38$_{\pm 0.39}$|49.82$_{\pm 0.54}$|43.15$_{\pm 0.93}$|42.82$_{\pm 0.27}$|49.79|
> |DeLoRA|63.49$_{\pm 0.32}$|51.40$_{\pm 0.28}$|45.53$_{\pm 0.73}$|42.72$_{\pm 0.29}$|50.78|
> |HiRA|63.62$_{\pm 0.33}$|51.17$_{\pm 0.26}$|45.12$_{\pm 0.61}$|42.89$_{\pm 0.20}$|50.70|
> |$S^2$-SAM|**63.73$_{\pm 0.38}$**|50.80$_{\pm 0.23}$|43.09$_{\pm 0.20}$|42.44$_{\pm 0.19}$|50.02|
> |**Bi-LoRA**|63.67$_{\pm 0.15}$|**51.53$_{\pm 0.33}$**|**46.12$_{\pm 0.89}$**|**43.45$_{\pm 0.15}$**|**51.19**|
>
> [1] MELoRA: Mini-Ensemble Low-Rank Adapters for Parameter-Efficient Fine-Tuning
>
> [2] DeLoRA: Decoupling Angles and Strength in Low-rank Adaptation
>
> [3] HiRA: Parameter-Efficient Hadamard High-Rank Adaptation for Large Language Models
>
> [4] A Single-Step, Sharpness-Aware Minimization is All You Need to Achieve Efficient and Accurate Sparse Training
>
> [5] LoRA-GA: Low-Rank Adaptation with Gradient Approximation

---

### Official Review · Reviewer_b4TB · 2025-11-01

**Soundness:** 3
**Presentation:** 3
**Contribution:** 3
**Rating:** 4
**Confidence:** 4

**Summary:**

The paper introduces a new method for LoRA training. In particular, the paper focuses on incorporating Sharpness-Aware Minimization (SAM) into LoRA training to improve the generalization performance of LoRA. The paper notes that direct naive application of SAM to LoRA limits the performance and incurs overhead. As such, the paper proposes to introduce independent perturbation via an auxiliary LoRA module, decoupling the sharpness optimization and task adaptation. As a result, the proposed method, Bi-LoRA, significantly reduces the training cost and prevents the perturbations from collapsing into the restricted optimization subspace. The experiemntal results demonstrate the strong performance of Bi-LoRA.

**Strengths:**

- The proposed idea of decoupling the perturbation from the optimization has clear motivations.
- The proposed idea is simple and effective.
- The proposed method does not bring significant overhead and yet brings performance improvement.

**Weaknesses:**

- The paper does not compare against other LoRA variants, such as PiSSA and DoRA on the entire benchmark.
- The paper notes that Bi-LoRA shares similar perspective with WSAM (Yue et al., 2023) that views SAM as regularization. However, the paper does not experimentally compare against WSAM.
- The paper does not provide theoretical justification as to how Bi-LoRA achieves better generalization compared to LoRA and LoRA-SAM.
- Bi-LoRA introduces additional hyperparameter that seems to need to be tuned for each dataset.

**Questions:**

- Could authors provide an intuitive explanation as to how the Bi-LoRA is able to give appropriate aligned perturbations, despited the fact that optimization and perturbation are decoupled?

---

> ### Author Response · Authors · 2025-11-21
> **Response to Reviewer b4TB (part 1)**
>
> ### R1.1. New Baselines.
>
> > W1: The paper does not compare against other LoRA variants, such as PiSSA and DoRA on the entire benchmark.
>
> > W2: The paper notes that Bi-LoRA shares similar perspective with WSAM (Yue et al., 2023) that views SAM as regularization. However, the paper does not experimentally compare against WSAM.
>
> Thanks for your suggestion. In our previous experiments, PiSSA and DoRA have been compared with Bi-LoRA on the CoLA dataset using T5-base in Section 4.5. Now we additionally evaluate them on the instruction-tuning tasks using Llama-3.1-8B, reported in Table R1.1. Due to the limited time, we cannot consider the entire benchmark. But hope this typical fine-tuning LLM task is sufficient to demonstrate the advantage of Bi-LoRA.
>
> Table R1.1: Instruction-tuning results on fine-tuning Llama 3.1-8B with LoRA, PiSSA, DoRA, WSAM and Bi-LoRA.
>
> | Method      | MMLU                   | DROP                   | HEval                  | BBH                    | Avg       |
> | ----------- | ---------------------- | ---------------------- | ---------------------- | ---------------------- | --------- |
> | LoRA        | 63.38$_{\pm 0.39}$     | 49.82$_{\pm 0.54}$     | 43.15$_{\pm 0.93}$     | 42.82$_{\pm 0.27}$     | 49.79     |
> | PiSSA       | 63.59$_{\pm 0.14}$     | 50.20$_{\pm 0.20}$     | 43.86$_{\pm 0.12}$     | 43.25$_{\pm 0.51}$     | 50.22     |
> | DoRA        | 63.58$_{\pm 0.22}$     | 50.53$_{\pm 0.10}$     | 44.10$_{\pm 0.73}$     | 42.98$_{\pm 0.79}$     | 50.30     |
> | WSAM        | 63.42$_{\pm 0.12}$     | 50.63$_{\pm 0.14}$     | 42.88$_{\pm 0.81}$     | 43.16$_{\pm 0.52}$     | 50.02     |
> | **Bi-LoRA** | **63.67$_{\pm 0.15}$** | **51.53$_{\pm 0.33}$** | **46.12$_{\pm 0.89}$** | **43.45$_{\pm 0.15}$** | **51.19** |
>
> ### R1.2. No theoretical justification for Bi-LoRA's better generalization results.
>
> > W3: The paper does not provide theoretical justification as to how Bi-LoRA achieves better generalization compared to LoRA and LoRA-SAM.
>
> Thanks for your question on our advantages over LoRA and LoRA-SAM.
>
> Compared to LoRA, Bi-LoRA follows the standard framework of generalization analysis for SAM, which is therefore not included here.
>
> Compared to LoRA-SAM, Bi-LoRA is more efficient: Bi-LoRA uses a single gradient step, which avoids LoRA-SAM's doubled cost. We have discussed this point.
>
> Regarding the advantage in generalization over LoRA-SAM, we admit that comprehensive theoretical results are lacking, due to an essential difficulty: how to theoretically characterize how the perturbation subspace’s dimensionality and scope relate to full-space flatness and generalization. Instead, we use discussions (see Lines 77-79 in the main text and R4.1 for Reviewer K3nD) and detailed experiments (see Tables 1, 2, as well as additional experiments requested by other reviewers) to convey our view that the decoupled form of Bi-LoRA leads to better generalization. We hope you could understand the theoretical difficulty, and we are glad to provide more empirical evidence in the discussion.

---

> > ### Author Response · Authors · 2025-11-21
> > **Response to Reviewer b4TB (part 2)**
> >
> > ### R1.3. Bi-LoRA introduces additional hyperparameters.
> >
> > > W4: Bi-LoRA introduces additional hyperparameters that seems to need to be tuned for each dataset.
> >
> > Thank you for raising the question about Bi-LoRA’s additional hyperparameters. Bi-LoRA introduces three extra hyperparameters beyond vanilla LoRA: (1) $\rho$ for perturbation, (2) auxiliary rank $r_2$ and (3) learning rate lr2 for the decoupled auxiliary module.
> >
> > The sensitivity of the LoRA ranks has already been discussed in Section 4.6 and Figure 6. We show that raising auxiliary rank $r_2$ from 2 to 8 consistently improves performance, with diminishing returns beyond that. Therefore, we recommend using $r_2=8$.
> >
> > Appendix H.2 (Table A3) reports results on CoLA, SST2, and GSM8K, where $\rho$ = 0.05 or 0.1 is typically optimal or near-optimal, and we now provide additional instruction-tuning experiments on Llama 3.1-8B (Table R1.3-a), which show the same pattern: $\rho= 0.05$  achieves the highest average score, with $\rho$ = 0.1 close, while $\rho$ = 0.01 and 0.2 underperform.
> >
> > And for step size of the auxiliary module lr2, Appendix O (Table A6) on CoLA and our new instruction-following results in Table R1.3-b both indicate that explicitly tuning η₂ yields only marginal changes: using the same learning rate for the primary and auxiliary modules is usually the best setting, while avoiding additional hyperparameter tuning. We conjecture that this insensitivity is partly due to the global norm clipping on the auxiliary module.
> >
> > Based on these observations, we suggest $r_2 = 8, \rho \in \lbrace0.05, 0.1\rbrace$, and lr1 = lr2 as practical defaults for Bi-LoRA in fine-tuning scenario.
> >
> > Table R1.3-a: Instruction-tuning results on fine-tuning Llama 3.1-8B with Bi-LoRA, tuning $\rho$.
> >
> > | $\rho$ | MMLU                   | DROP                   | HEval                  | BBH                    | Avg       |
> > | ------ | ---------------------- | ---------------------- | ---------------------- | ---------------------- | --------- |
> > | 0.01   | 63.47$_{\pm 0.25}$     | 50.16$_{\pm 0.20}$     | 43.09$_{\pm 0.41}$     | 43.08$_{\pm 0.30}$     | 49.95     |
> > | 0.05   | 63.67$_{\pm 0.15}$     | **51.53$_{\pm 0.33}$** | **46.12$_{\pm 0.89}$** | 43.45$_{\pm 0.15}$     | **51.19** |
> > | 0.1    | 63.53$_{\pm 0.11}$     | 50.64$_{\pm 0.12}$     | 45.32$_{\pm 0.81}$     | **43.89$_{\pm 0.06}$** | 50.85     |
> > | 0.2    | **63.76$_{\pm 0.30}$** | 50.34$_{\pm 0.23}$     | 44.92$_{\pm 1.08}$     | 43.61$_{\pm 0.18}$     | 50.66     |
> >
> > Table R1.3-b: Instruction-tuning results on fine-tuning Llama 3.1-8B with Bi-LoRA, tuning lr2 with lr1=3e-4.
> >
> > | (lr1, lr2)   | MMLU                   | DROP                   | HEval                  | BBH                    | Avg       |
> > | ------------ | ---------------------- | ---------------------- | ---------------------- | ---------------------- | --------- |
> > | (3e-4, 1e-4) | **63.89$_{\pm 0.25}$** | 51.05$_{\pm 0.13}$     | 45.12$_{\pm 0.35}$     | 42.84$_{\pm 0.15}$     | 50.73     |
> > | (3e-4, 3e-4) | 63.67$_{\pm 0.15}$     | **51.53$_{\pm 0.33}$** | **46.12$_{\pm 0.89}$** | **43.45$_{\pm 0.15}$** | **51.19** |
> > | (3e-4, 5e-4) | 63.47$_{\pm 0.35}$     | 51.23$_{\pm 0.24}$     | 45.93$_{\pm 0.73}$     | 43.05$_{\pm 0.12}$     | 50.92     |
> > | (3e-4, 1e-3) | 63.40$_{\pm 0.17}$     | 51.32$_{\pm 0.22}$     | 44.51$_{\pm 0.93}$     | 42.96$_{\pm 0.23}$     | 50.55     |
> >
> > ### R1.4. How the Bi-LoRA is able to give appropriate alignment.
> >
> > > Q1: Could authors provide an intuitive explanation as to how the Bi-LoRA is able to give appropriate aligned perturbations, despited the fact that optimization and perturbation are decoupled?
> >
> > Thank you for raising this insightful question. Bi-LoRA optimizes adversarial perturbations by introducing an auxiliary LoRA module, $(B_2, A_2)$, resulting in the parameterization $W = W_0 + B_1A_1 + B_2A_2$. The auxiliary module is continuously updated using gradient ascent (with norm-$\rho$ ball constraint) in parallel with gradient descent on the primary LoRA module, $(B_1, A_1)$. The intuition behind this approach is that adversarial perturbations across different steps are "continuous": the optimal adversarial perturbation at iteration $t$ is likely to be close to the optimal perturbation at iteration $t+1$, given that the weight differences between the continuous steps are small under LoRA optimization. This means we can obtain the adversarial perturbation at step $t+1$ with a "slight" adjustment based on the perturbation at $t$. As a result, this method can effectively optimize the sharpness while maintaining the same training efficiency as regular LoRA training.

---

> > > ### Comment · Reviewer_b4TB · 2025-11-28
> > >
> > > The reviewer appreciates the authors' rebuttal that has addressed the concerns. Therefore, the rating has been updated accordingly.

---

> > > > ### Author Response · Authors · 2025-11-28
> > > > **Official Comment by Authors**
> > > >
> > > > Thank you very much for your kind follow-up and for taking the time to revisit our submission. We are glad that our responses have addressed your concerns. We sincerely appreciate your updated rating.

---

### Author Response · Authors · 2025-11-21
**General Response to Reviewers**

We thank all reviewers for taking time to carefully read our paper and for their detailed and constructive feedback. We are encouraged to see that reviewers find the strengths of Bi-LoRA: clear motivation (b4TB, 6yuC); efficiency improvements (b4TB, bc19, 6yuC, K3nD); and a simple yet effective design (b4TB, bc19, 6yuC, K3nD). In this rebuttal, we hope to address all raised concerns through additional experiments and further clarifications.

Experiments:

- We add WSAM, $S^2$-SAM, PiSSA, DoRA, HiRA, and DeLoRA as new baselines on the instruction following tasks with Llama 3.1-8B. (R1.1 for Reviewer b4TB, R2.3 for Reviewer bc19)
- We conduct new ablations over key hyperparameters, including the perturbation radius $\rho$, the auxiliary rank $r_2$, and the auxiliary step size $\text{lr}_2$. (R1.3 for Reviewer b4TB, R2.3 for Reviewer bc19, R3.3 for Reviewer 6yuC)
- We further ablate the main components of Bi-LoRA to identify the source of its improvements. In particular, we study the effects of the clipping form, the adversarial mechanism, and the presence of the auxiliary modules, and find that both global clipping and the additional adversarial branch are indispensable for the higher performance of Bi-LoRA. (R2.1 for Reviewer bc19, R3.2 for Reviewer 6yuC)


Clarifications:

- We further clarify the motivation and contributions of Bi-LoRA. Our main contribution is to retain the efficiency of LoRA-style fine-tuning while addressing a key limitation of LoRA-SAM: it perturbs only a restricted low-rank subspace, which may constrain its potential for generalization. (R1.2 for Reviewer b4TB, R2.2 for Reviewer bc19, R4.2 for Reviewer K3nD)
- We explain that, even when adopting a LoRA parameterization, it remains important to explore perturbations within a broader effective space. Bi-LoRA is specifically designed to expand this exploration of perturbations without compromising efficiency. (R4.1 for Reviewer K3nD)
- We clarify why the auxiliary LoRA module converges rapidly only after the main LoRA branch has plateaued. (R4.5 for Reviewer K3nD)
- We provide an intuitive explanation for why Bi-LoRA, despite its decoupled formulation, still yields aligned perturbations. (R1.4 for Reviewer b4TB, R4.3 for Reviewer K3nD)
- We ablate and justify our choice of using a global clipping strategy. (R3.4 for Reviewer 6yuC)


We would like to once again sincerely thank the reviewers for their insightful comments and suggestions. We are currently preparing the revised PDF and look forward to continuing the discussion with the reviewers. All clarifications, new results, and any consensus reached during the discussion phase will be carefully incorporated into the revised version.

---

### Author Response · Authors · 2025-11-27
**General Response**

Dear Program Chairs, Senior Area Chairs, Area Chairs, and Reviewers,

We again thank you for your thoughtful feedback and careful evaluation of our work. We have updated the revised PDF, in which we have made the following changes and additions in response to the reviews:

**Experiments**
- Baselines: $S^2$-SAM, PiSSA, DoRA, HiRA, and DeLoRA. (R1.1 for Reviewer b4TB, R2.3 for Reviewer bc19 | Appendix H, Table A1)
- Ablations and hyperparameter sensitivity. (R1.3 for Reviewer b4TB, R2.3 for Reviewer bc19, R3.3 for Reviewer 6yuC | Section 4.6, Tables 5, 6, Figure 6)

**Clarification**
- Fixing the overclaim. (R3.1 for Reviewer 6yuC | Section 3  Lines 252-253)
- Discussion on Proposition 2. (R1.2 for Reviewer b4TB, R2.2 for Reviewer bc19, R4.3 for Reviewer K3nD | Appendix F.1)
- Discussion on Proposition 3. (R1.4 for Reviewer b4TB, R4.3 for Reviewer K3nD | Appendix F.2)
- Reason of using global clipping in Bi-LoRA. (R3.4 for Reviewer 6yuC | Appendix G)

We are deeply grateful for the reviewers’ insightful comments, which have helped us improve the clarity and completeness of the manuscript. We welcome any further discussions.

---

### Author Response · Authors · 2025-12-02
**Summary to ACs**

Dear Program Chairs, Senior Area Chairs, Area Chairs, and Reviewers,

We thank all reviewers for carefully evaluating our submission and their insightful feedback.

Bi-LoRA offers a simple dual-LoRA design that efficiently realizes SAM-style generalization improvement, which the reviewers have generally acknowledged. During the discussion, Reviewer b4TB indicated that "the rebuttal has addressed the concerns and therefore **the rating has been updated accordingly**". Reviewers' concerns focused on:
-  **Alignment of the previous gradient (Proposition 2).** Reviewer b4TB asks for an intuitive explanation of how this alignment works, and we provide the intuition that adversarial perturbations are approximately continuous across optimization steps. Reviewer K3nD similarly questions whether aligning with the previous gradient is valid across iterations, and we have therefore clarified this approximation and strengthened its connections to prior work.
- **The gap between Bi-LoRA's adversarial direction and the ideal direction (Proposition 3)**. Reviewers bc19 and K3nD do not dispute the correspondence between Bi-LoRA and SAM in the ideal setting, but they raise the concern that using a one-step update to construct the adversarial direction introduces a gap that may affect generalization. Reviewer b4TB also asks for a theoretical account of Bi-LoRA’s generalization gains. In response to these concerns, we emphasize that our approximation is efficiency-oriented, and our experiments consistently show that Bi-LoRA still improves generalization in practice despite this gap.

For experiments, Reviewers b4TB, bc19, and 6yuC provide valuable suggestions. We have added PiSSA, HiRA, DeLoRA, WSAM, and $S^2$-SAM as new baselines, and performed extensive ablations and hyperparameter sensitivity studies to quantify the source of Bi-LoRA’s gains and provide practical tuning guidance.

**Note on revision**: All updates, including the above clarifications and new experiments, have been highlighted in blue in the revised PDF for easy tracking.

We sincerely hope this summary helps facilitate your evaluation by clearly demonstrating how the main concerns have been addressed. We are sincerely grateful to the reviewers and ACs for the time and thoughtful consideration devoted to our work. Given the exceptional reviewing circumstances this year, we are especially appreciative of the additional effort you have invested in managing the review process.

Best regards,

Authors of Submission 11559

---

### Meta-Review · Area_Chair_mTte · 2025-12-16

**Summary:**

Initial scores were 8, 8, 4, 4.
Reviewers acknowledged the simple yet elegant design, strong efficiency gains (near-LoRA cost while approximating SAM), and comprehensive empirical validation across LLMs and diffusion models. Main concerns raised were: (1) missing baseline comparisons (PiSSA, DoRA, WSAM) and incomplete ablations isolating the adversarial ascent contribution, (2) theoretical gaps between idealized propositions and the single-step practical algorithm, (3) hyperparameter tuning requirements and (4) motivation clarity regarding why broader perturbation space exploration is necessary and whether Bi-LoRA actually achieves it given its own subspace constraints.

**Reviewer Concerns:**

**Addressed:**
- **Missing baselines (b4TB, bc19):** Authors added PiSSA, DoRA, WSAM comparisons on Llama 3.1-8B instruction tuning. Additional single-pass SAM variants (\delta-SAM) and LoRA variants (DeLoRA, HiRA) also compared.
- **Ablation isolating adversarial ascent effect (bc19):** Dual-LoRA control experiment (both modules descent, same ranks and clipping) shows marginal gain over LoRA (49.86 vs. 49.79), confirming improvements stem from adversarial ascent mechanism, not simply added capacity.
- **Hyperparameter sensitivity (b4TB, bc19, 6yuC):** Exended sensitivity analyses provided for various hyperparam combinations along with recommend practical defaults.
- **Clipping strategy ablations (bc19, 6yuC):** Authors compared global vs. per-layer vs. no clipping, demonstrating global clipping significantly outperforms and justifying global norm as faithful SAM analogue.


**Outstanding:**
- **Theoretical gap (bc19, K3nD):** Authors acknowledge the gap between idealized Proposition 3 (perfect inner maximization) and practical single-step ascent. Comprehensive theory for the approximate algorithm remains future work.
- **Limited absolute gains (K3nD):** Improvements are modest but understandable given strong tuned baselines. Authors note limited headroom for improvement when comparing against well-optimized LoRA.

**Reviewer Scores:**

- **Reviewer b4TB (initial: 4):** Reviewer explicitly confirmed updating rating (probably to 6) after rebuttal: "The reviewer appreciates the authors' rebuttal that has addressed the concerns. Therefore, the rating has been updated accordingly." All major concerns (missing PiSSA/DoRA/WSAM baselines, hyperparameter tuning, alignment intuition) were directly addressed with new experiments.

- **Reviewer bc19 (initial: 8):** Would likely remain at 8. Reviewer gave high initial score and all specific concerns were addressed in rebuttal. Outstanding theoretical gap is acknowledged as difficult future work.

- **Reviewer 6yuC (initial: 8):** Would likely remain at 8. Reviewer rated presentation as excellent and contribution as good. Ablation requests were thoroughly addressed with Dual-LoRA control and clipping experiments. High confidence 5/5

- **Reviewer K3nD (initial: 4):** May increase to 6. Authors provided detailed clarifications on motivation (why broader exploration, how Bi-LoRA achieves it via decoupling), theoretical propositions' roles (ideal vs. practical), and auxiliary convergence behavior. While theoretical gaps remain, empirical evidence and intuitive explanations address core concerns about motivation and mechanism.

---

### Decision · Program_Chairs · 2026-01-26

Accept (Poster)